# Targeted chromosomal *Escherichia coli:dnaB* exterior surface residues regulate DNA helicase behavior to maintain genomic stability and organismal fitness

Megan S. Behrmann[1], Himasha M. Perera[1], Joy M. Hoang[1¤], Trisha A. Venkat[1], Bryan J. Visser[2], David Bates[2], Michael A. Trakselis[1]*

1 Department of Chemistry and Biochemistry, Baylor University, Waco, Texas, United States of America, 2 Department of Molecular and Human Genetics, Baylor College of Medicine, Houston, Texas, United States of America

¤ Current address: Baylor College of Medicine, Houston Texas
* michael_trakselis@baylor.edu

**Data Availability Statement:** All sequencing data generated in this study have been deposited to the

## Abstract

Helicase regulation involves modulation of unwinding speed to maintain coordination of DNA replication fork activities and is vital for replisome progression. Currently, mechanisms for helicase regulation that involve interactions with both DNA strands through a steric exclusion and wrapping (SEW) model and conformational shifts between dilated and constricted states have been examined *in vitro*. To better understand the mechanism and cellular impact of helicase regulation, we used CRISPR-Cas9 genome editing to study four previously identified SEW-deficient mutants of the bacterial replicative helicase DnaB. We discovered that these four SEW mutations stabilize constricted states, with more fully constricted mutants having a generally greater impact on genomic stress, suggesting a dynamic model for helicase regulation that involves both excluded strand interactions and conformational states. These *dnaB* mutations result in increased chromosome complexities, less stable genomes, and ultimately less viable and fit strains. Specifically, *dnaB:mut* strains present with increased mutational frequencies without significantly inducing SOS, consistent with leaving single-strand gaps in the genome during replication that are subsequently filled with lower fidelity. This work explores the genomic impacts of helicase dysregulation *in vivo*, supporting a combined dynamic regulatory mechanism involving a spectrum of DnaB conformational changes and relates current mechanistic understanding to functional helicase behavior at the replication fork.

## Author summary

DNA replication is a vital biological process, and the proteins involved are structurally and functionally conserved across all domains of life. As our fundamental knowledge of genes and genetics grows, so does our awareness of links between acquired genetic

Sequence Read Archive, https://www.ncbi.nlm.nih.
gov/sra (accession no. PRJNA773110).

**Funding:** This work was funded by NSF MCB
(NSF1613534 and NSF2105167 to M.A.T.),
supported by Baylor University (M.A.T) and the
NIH (R01GM135368 to D.B.). The funders had no
role in study design, data collection and analysis,
decision to publish, or preparation of the
manuscript.

**Competing interests:** The authors have declared
that no competing interests exist.

mutations and disease. Understanding how genetic material is replicated accurately and efficiently and with high fidelity is the foundation to identifying and solving genome-based diseases. *E. coli* are model organisms, containing core replisome proteins, but lack the complexity of the human replication system, making them ideal for investigating conserved replisome behaviors. The helicase enzyme acts at the forefront of the replication fork to unwind the DNA helix and has also been shown to help coordinate other replisome functions. In this study, we examined specific mutations in the helicase that have been shown to regulate its conformation and speed of unwinding. We investigate how these mutations impact the growth, fitness, and cellular morphology of bacteria with the goal of understanding how helicase regulation mechanisms affect an organism's ability to survive and maintain a stable genome.

## Introduction

Faithful and efficient DNA replication is a fundamental life process that is the result of complex interactions between a diverse collection of enzymes. Proximal to this process is the DNA helicase enzyme, a hexameric protein ensemble that separates double-stranded DNA (dsDNA) for synthesis and coordinates replicative actions. The functional mechanism for unwinding by this toroidal-shaped enzyme is well studied, however the method by which the helicase is regulated remains unresolved [1,2]. It is known that the helicase unwinds by steric exclusion (SE); a mechanism that involves encircling and translocating on one strand with a particular polarity, while the complementary strand is excluded from the central channel [2,3]. The excluded strand has also been implicated in helicase regulation [4–9] by interacting electrostatically on the exterior of many different helicases from multiple organisms to control unwinding speed, establishing the steric exclusion and wrapping (SEW) model for unwinding [10–13]. Other mechanisms for helicase regulation have been proposed to modulate both the speed of unwinding, coupled unwinding and synthesis, and coordinated priming [14–19], but little has been done to examine the cellular consequences of helicase dysregulation *in vivo*.

*Escherichia coli* (*E. coli*) DnaB is a homohexameric superfamily 4 (SF4) helicase that encircles and translocates along the lagging strand in the 5'-3' direction [20,21]. Not only are SF4 helicases well characterized, but *E. coli* is a well-tested model organism, making this an ideal system for investigating replisome mechanics *in vivo*. The SE model provides a mechanism for single-strand DNA (ssDNA) translocation and prevents immediate reannealing of the DNA strands during unwinding; while the SEW model adds a dynamic interaction with the excluded strand to modulate enzyme activity. Biochemical analyses of DnaB indicated that a stable interaction with the excluded strand restricts unwinding, likely acting as a brake to slow helicase progression [4]. Site-specific external SEW mutations of DnaB resulted in 20 to 50-fold increases in DNA unwinding *in vitro*. This regulation may be important *in vivo* to limit separation of DnaB from the replisome, which may occur during Okazaki fragment priming or during helicase-polymerase decoupling [22–25]. Functionally, this would promote coordinated helicase-polymerase coupled DNA replication and aid in preventing ssDNA buildup from uncontrolled unwinding to limit chromosomal breaks [26].

In addition to direct interactions with DNA strands, the DnaB hexamer also undergoes a large conformational change upon interaction with specific replisome components: clamp-loader complex (CLC) (specifically tau), primase (DnaG), and the helicase loader (DnaC). At least two distinct conformations of the DnaB helicase have been observed: a dilated state that favors interaction with DnaG [27] and was shown to unwind similar to wild-type (WT) DnaB

*in vitro* [14] and a constricted state that favors interaction with DnaC [28] and resulted in rapid unwinding relative to WT that was further stimulated by addition of tau [14]. These conformational dynamics likely also affect the binding affinity of the excluded strand, ultimately controlling the unwinding speed for the replisome.

In this report, we confirm that the faster unwinding SEW DnaB mutants also stabilize a constricted helicase conformation, indicating that both excluded strand access to the exterior surface and the conformational state of the helicase contribute to the structure/function mechanism for regulating unwinding speed. Identical site-specific genomic mutations of *E. coli dnaB* were engineered using CRISPR-Cas9 editing to investigate the effects of helicase regulation on cellular fitness and overall genomic stability. Generally, *dnaB* mutants grew slower, were outcompeted by the parental strain, and displayed a filamentous cell phenotype, indicating more genomic and cellular stress. Fluorescence activated cell sorting (FACS) and quantitative PCR (qPCR) analysis indicated higher chromosome numbers and increased *ori:ter* ratios indicating dysregulation of replication processes that may be caused by altered DnaB loading for initiation for some of the mutants. Generally correlating with the hexameric conformation, the *dnaB* mutant strains had a spectrum of genomic instabilities, including increased mutagenesis and prevalent free 3' ends, while not requiring SOS induction for growth and survival, consistent with leaving small gaps in DNA during replication that are filled in an error prone manner. This work has important implications regarding the impact and importance of regulation of replisome speed on genomic stability, replication efficiency, and cellular fitness.

## Results

### DnaB SEW mutants favor a constricted conformation

The DnaB hexamer is known to adopt both constricted and dilated conformations (**S1 Fig**). DnaC stabilizes a constricted cracked conformation (lock washer) for loading [28]; the dilated conformation favors DnaG recruitment for priming [27]; and the τ subunit of the CLC stimulates the closed constricted state to couple rapid DNA synthesis with unwinding [14]. The constricted conformation is more efficient for translocation and unwinding but is unable to transverse over duplex DNA. To determine whether our DnaB SEW protein mutants enforce one conformation over the other to explain the increased DNA unwinding rates [4], we utilized a duplexed fluorescence translocation fork assay, similar to that described previously (**Fig 1**) [14].

Purified WT and mutant DnaB enzymes were incubated with a forked Cy3-fluorescently labeled and black hole quencher (BHQ) substrate, containing a duplex region prior to a 3' fork displaced strand (**Fig 1B**). DnaB is unable to unwind a 5' single arm substrate (**S2 Fig**) [29], and so, it must translocate over the duplex region to separate the Cy3-reporter strand. Only DnaB hexamers that can adopt a dilated state or fluctuate between conformational states can move over the duplex to unwind the Cy3-reporter strand from the BHQ strand, resulting in an increase in the fluorescent signal upon addition of ATP (**Fig 1B**) [14]. WT DnaB is able to freely switch between conformations and unwound 43% of the total DNA substrate (**Fig 1A**). Both DnaB (K180A) and DnaB (R328/9A) have essentially no increases in fluorescence over time, unwinding only ~5%, similar to the no ATP negative control. In a previous report, both K180A and R328/9A had 20-fold increases in unwinding rates on traditional forked substrates [4]; however, in this duplex translocation assay, both K180A and R328/9A mutants are unable to translocate over duplex DNA and therefore likely maintain a static fully constricted conformation. DnaB(R74A) and DnaB(R164A) both show moderate increases in fluorescence with 18% and 15% Cy3 strand unwound, respectively. This demonstrates that although R74A and R164A can switch to a dilated state, these mutations also likely shift the equilibria towards the

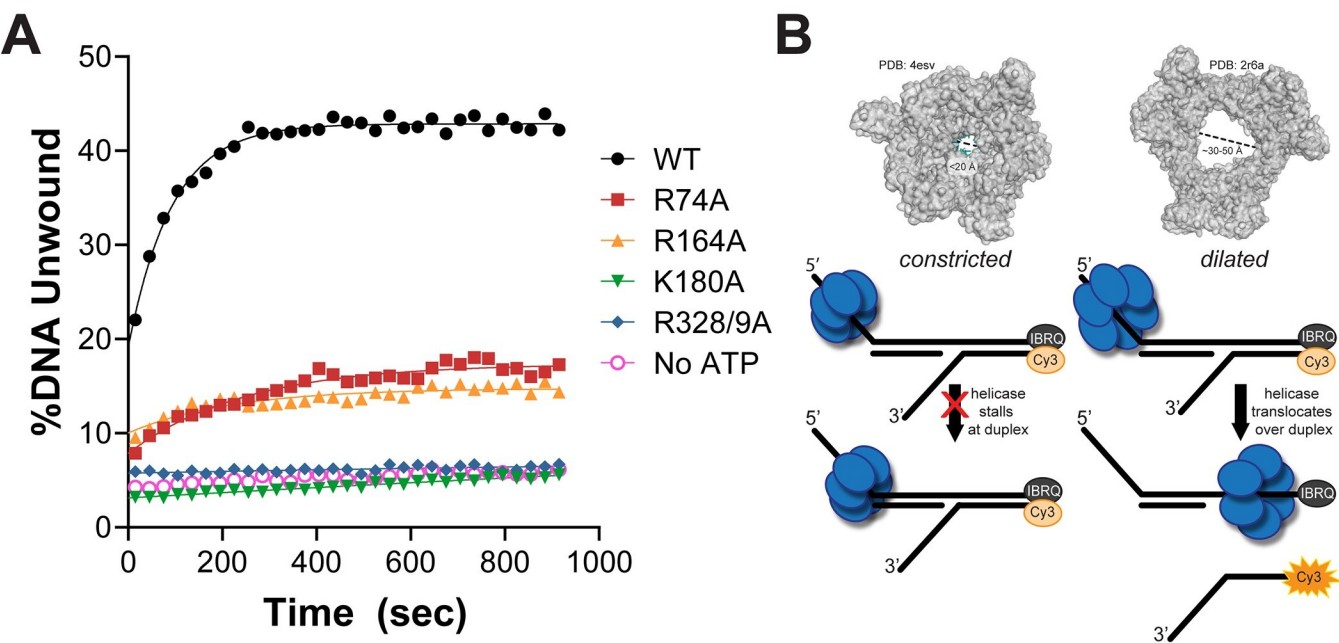

**Fig 1. DnaB mutants have impaired ability to translocate over duplex DNA consistent with a more constricted state.** (**A**) A plot comparing the amount of Cy3-DNA unwound for each mutant as a function of time for this duplex translocation assay. Average of n = 7 replicates shown. Data is fit to **Eq 1**. Error bars represent ± standard deviation (SD) and are within symbols where not visible. Negative control is WT DnaB with all the reaction components excluding ATP (No ATP, purple). Rates of unwinding are $10.7 \pm 0.5 \times 10^{-3}$ s$^{-1}$ for WT (black ●), $4.0 \pm 0.4 \times 10^{-3}$ s$^{-1}$ for R74A (red ■), $4.5 \pm 0.9 \times 10^{-3}$ s$^{-1}$ for R164A (orange ▲), $0.0 \pm 0.3 \times 10^{-3}$ s$^{-1}$ for K180A (green ▼), $0.0 \pm 1.3 \times 10^{-3}$ s$^{-1}$ for R328/9A (blue ◆), and $0.0 \pm 0.7 \times 10^{-3}$ s$^{-1}$ for no ATP control (pink ○). The symbols and colors for the DnaB or *dnaB* mutants are consistent throughout. (**B**) Crystal structures of a constricted DnaB helicase bound to ssDNA (*G. stearothermophilus*, PDBD: 4esv), corresponding dilated *Gst*DnaB helicase (PDBID: 2r6a), and a schematic of the substrate design for duplex translocation assay liberating the Cy3 strand for increased fluorescence. Controls showing no unwinding of the short duplex oligo and unwinding of the 3' flap oligo are provided in **S2 Fig**.

constricted state, consistent with their 3-6-fold faster unwinding of traditional forks [4,14] but can still fluctuate somewhat to a dilated conformation. These mutants, (R74A and R164A), present states more moderately constricted and intermediate than previously seen.

### *In vivo dnaB* mutations limit growth and generate stress

To determine whether faster *in vitro* DNA unwinding with preferential constricted conformations for DnaB (R74A, R164A, K180A, and R328/9A) have detrimental effects on replication speed, genomic stability, and organismal fitness, we created site-specific genomic *dnaB* point mutations using CRISPR-Cas9 editing (**S3 Fig**). Several successfully edited *dnaB* colonies were obtained for each mutant with high frequencies ranging from 42–90%. Mass doubling times for all *dnaB* mutants was monitored at $OD_{600}$ in rich media and compared to the parental strain using a plate reader maintained at 32°C (to prevent induction of λ red genes). Although the doubling times are significantly reduced by growth at 32 °C and restricted agitation in 96-well plates, the assay provided a convenient, controlled, and quantitative method to compare relative growth rates between *dnaB:mut* strains. The absolute growth rate for most of the *dnaB* mutants was significantly decreased compared to the parental strain with the exception of *dnaB:R74A* (**Fig 2A**). Interestingly, within the mutant *dnaB* strains, there is variability in the fitted exponential growth curves. *dnaB:WT* and *dnaB:R74A* increase in density during exponential growth phase at a rate of approximately $2.9 \times 10^{-3}$ min$^{-1}$. *dnaB:K180A* has a growth rate of $1.4 \times 10^{-3}$ min$^{-1}$, half that of *dnaB:WT* and the slowest of all the mutants (**Fig 2B**). *dnaB:R164A* increases at a rate of $2.0 \times 10^{-3}$ min$^{-1}$, nearly 1.5 times slower than the parental

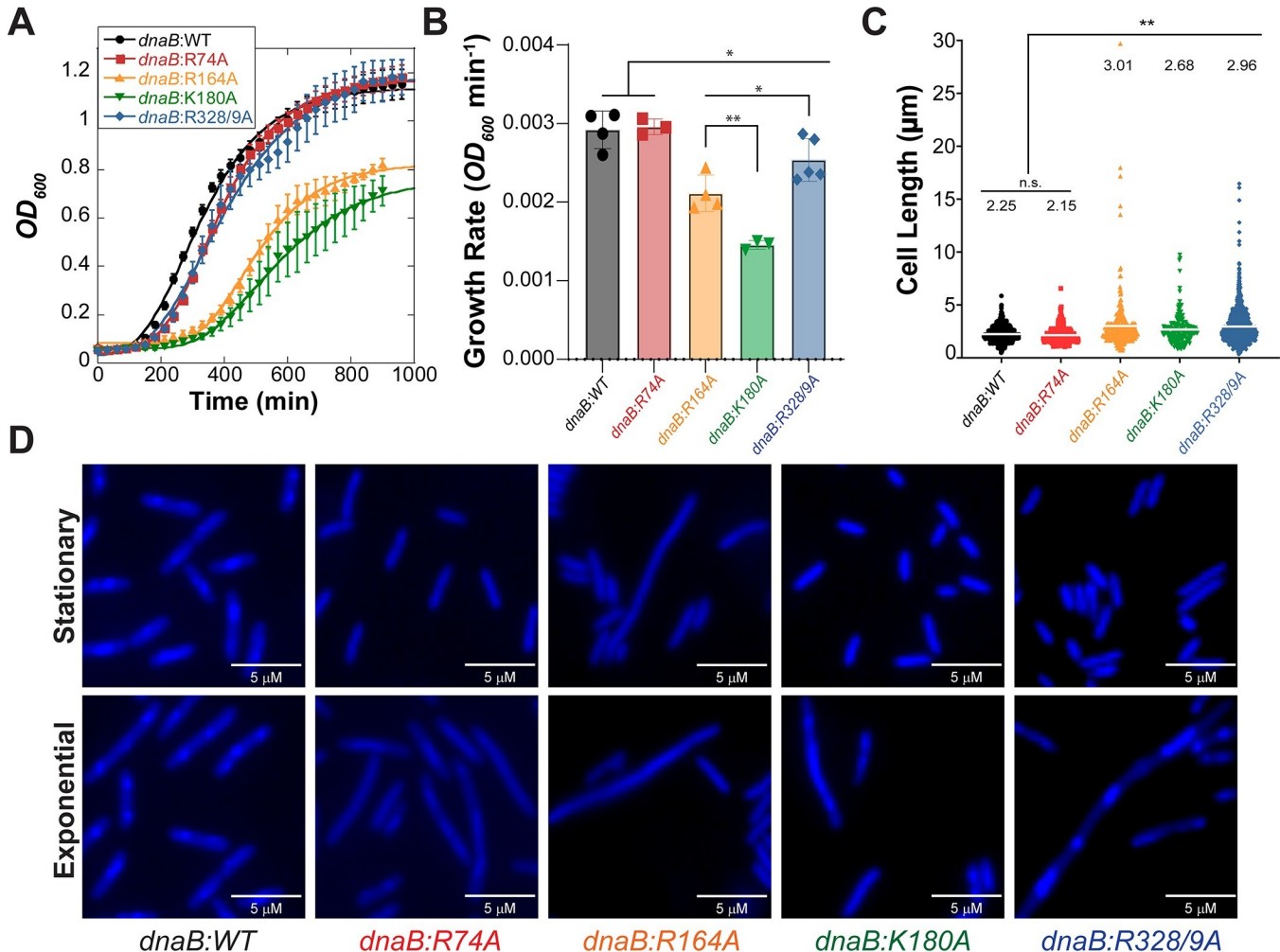

**Fig 2. Growth and visualization of cell morphology of *dnaB:mut* and *WT* strains.** (**A**) Growth curves of indicated *E. coli* strains grown at 32 °C are fitted to **Eq 2**. Data plotted is the mean for three trials of three technical replicates each (n = 9), and error bars represent ± SD (**B**) The averaged absolute growth rate for each strain is plotted for comparison. Individual data is presented with open circles. Black bars indicate statistically significant differences, where p-values are * < 0.05 and ** < 0.01. (**C**) The cell lengths were measured by blinded visual quantification for n ≥ 200 cells, and the average cell length is reported above the data points (grey bar) for each sample. Statistically significant differences are indicated, where p-values are ** < 0.01. n.s. is not significant. (**D**) Stationary and exponential growth cells are stained with DAPI and imaged using epifluorescence microscopy. Images shown are representative of the population observed. Wider views are provided in **S4 Fig** along with the quantification of stationary phase cell lengths.

strain. *dnaB:R328/9A* has a growth rate 2.5 x $10^{-3}$ min$^{-1}$, similar but still significantly slower than *dnaB:WT*.

Of course, growth rates are commonly and conveniently measured using absorbance, but this relies on a uniform cell size between conditions or strains for accurate measurements. A reduction in overall growth rate may indicate that these helicase mutations are causing genomic or cellular stress, which in bacteria is often typified by cellular elongation. To investigate this, we exposed exponentially growing samples to DAPI, imaged using an epifluorescent light microscope, measured the cell length, and analyzed these strains blindly (**Fig 2C**). All mutants, except *dnaB:R74A*, had significantly elongated cells during exponential growth phases relative to *dnaB:WT*, which had an average cell size of 2.25 ± 0.02 μm with sizes ranging up to 5 and 6 μm. *dnaB:R164A* and *dnaB:R328/9A* strains averaged 3.01 ± 0.14 and 2.96 ± 0.14 μm per cell, with *dnaB:R328/9A* having the largest population of longer cells (>5 μm) and cells ranging to

over 15 μm in length. *dnaB*:*R164A* had the longest cells reaching 20 μm in length. *dnaB*:*K180A* had an average cell length of 2.68 ± 0.10 μm with a maximum recorded size of 10 μm. Representative images of cell populations show visual increases in the filamented cell population, with inlays highlighting representative groups (**Fig 2D**). Additionally, large aggregates of filamented cells were observed for *dnaB*:*K180A* and *dnaB*:*R328/9A*. Actively dividing cells can clearly be seen in all the exponentially growing cultures, including the parental strain, and care was taken not to confuse them with filamenting cells.

We also examined stationary phase samples of the mutants (**S4 Fig**), and found that most strains maintained an average of about 2 μm in cell length with a maximum length of 5.1 μm, with the exception of *dnaB*:*R164A*. The *dnaB*:*R164A* mutant strain had had an average cell length of 2.6 ± 0.9 μm and a maximum cell length of 10.6 μm and was significantly different than WT.

## SEW is important for strain fitness and survival

To understand how the differences in growth rate and cell morphology affected overall strain fitness, we utilized a colorimetric direct growth competition assay [30–33]. The redox indicator triphenyl tetrazolium chloride (TTC) is colorless but becomes deep red when reduced [34]. Phage transduction was used to generate a knockout of the promotor for arabinose isomerase (MSB1: *HME63:ΔaraBAD*), inhibiting hydrolysis of arabinose [35]. When grown on agar plates containing both TTC and the reducing sugar arabinose, MSB1 produces dark red colonies, allowing for differentiation of a mixed population of the control strain (*ara⁻*) and a test strain (*ara⁺*) based on color (**Fig 3A**).

Co-culturing MSB1 with *dnaB*:*WT* (or any of the *dnaB* mutant stains) allowed the monitoring of the competitive growth and fitness in a red:white competition assay. As a control, the mixed population of MSB1(red):*dnaB*:*WT*(white) remained constant over 48 hours (**Fig 3B and 3C**). However, each of the *dnaB* mutant strains declined in population to near zero in favor of MSB1 within 48 hours. Using the rate of decline overtime, we calculated the selection rate of each strain (*Eq 4*) compared to MSB1 [36]. This is an inverse measure of strain fitness, with zero indicating equal fitness to the control strain, as is the case for competition with *dnaB*:*WT* (**Fig 3D**). All of the *dnaB* mutant strains were significantly outcompeted by MSB1 demonstrating a severely decreased strain efficacy. This includes *dnaB*:*R74A*, which competed poorly against the control strain with a selection rate of -1.34 ± 0.14 despite exhibiting normal exponential growth in **Fig 2B**. *dnaB*:*R328/9A* had the most competitive selection rate of the mutants with a value of -0.74 ± 0.16, followed by *dnaB*:*R164A* with a -1.13 ± 0.27 selection rate relative to MSB1. *dnaB*:*K180A* was the least competitive with a selection rate of -1.6 ± 0.27, correlating with its slower growth rate in **Fig 2B**.

## Helicase dysregulation increased chromosome complexity

To further elucidate how our helicase mutations affect replication cycles, quantitative fluorescence activated cell sorting (FACS) was performed on exponentially growing cells utilizing rifampicin and cephalexin to inhibit replication initiation and cell division and allowing for the completion (or 'run-out') of DNA synthesis. Resulting DNA histograms indicate DNA content with major peaks at common whole chromosomes (**Fig 4A**) and intermediate values representing incomplete run-out, likely resulting from inefficient replication elongation or DNA damage (**S5 and S6 Figs**) [37,38]. Relative to the parental strain, all mutants had marked increases in chromosomes, correlating with the elongated and filamented cells observed by microscopy for every strain except for *dnaB*:*R74A* (**Fig 2C**). Like most actively replicating bacteria, *dnaB*:*WT* had major peaks representing 2 and 4 chromosome integers during this 4 hour

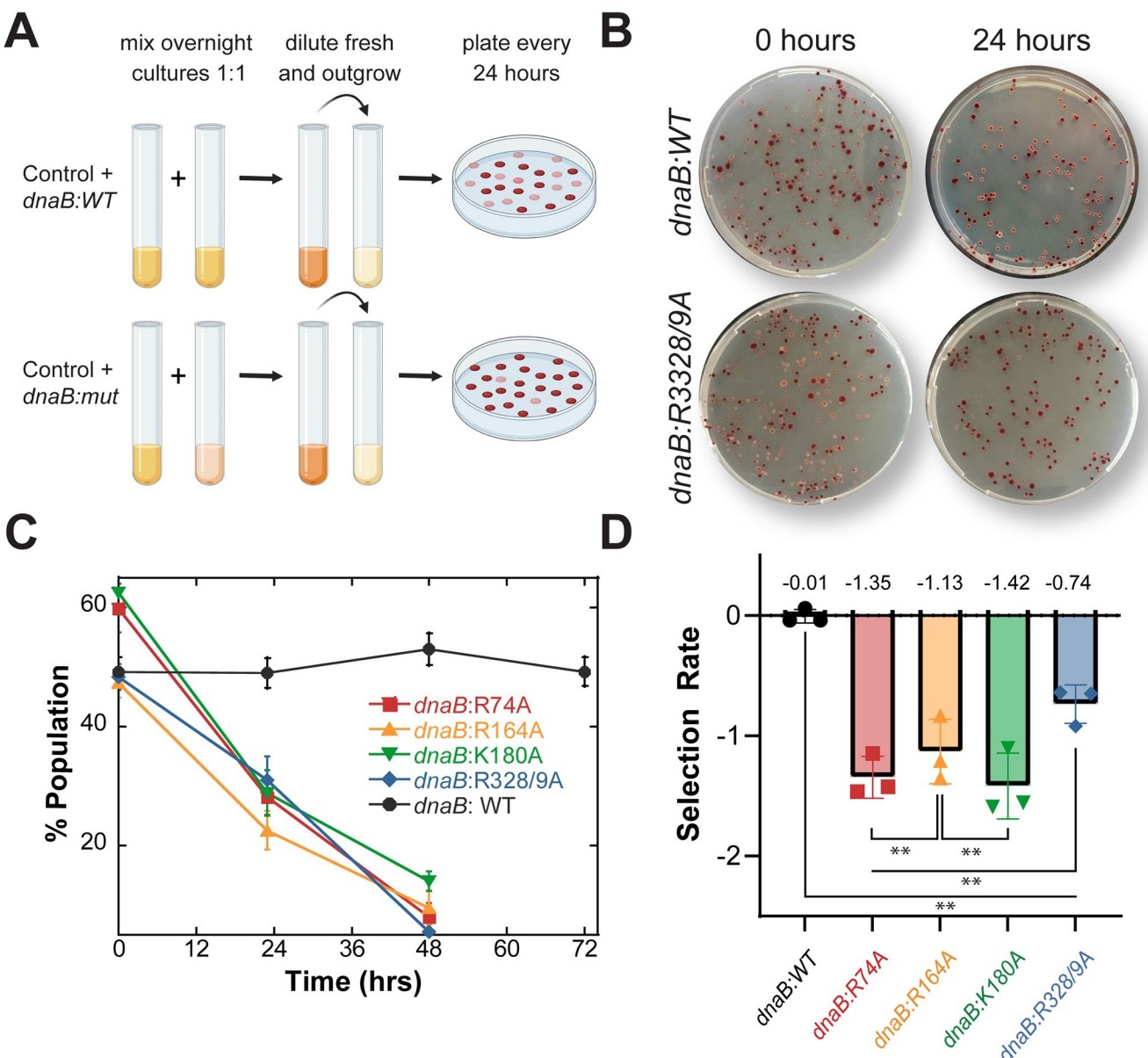

**Fig 3. *dnaB* mutations reduce strain fitness.** (**A**) Schematic for a red-white bacterial competition assay, where strains are color coded for colorimetric differentiation over time. Red is the control strain MSB1, and white are the test strains. (**B**) Representative plates showing changes in population overtime for the equally mixed populations MSB1 versus *dnaB:WT* or *dnaB:R328/9A* at 0 and 24 hours. (**C**) Changes in population of mixed cultures are plotted over time for n = 3 biological replicates with error bars representing the standard deviation. (**D**) The mean selection rate for each strain based on the population change during the first 24 hours (**Eq 4**). Individual data is presented with open circles. Black bars indicate statistically significant differences. Selection rate is a relative determination of strain fitness compared to the control strain, MSB1. Error bars represent ± SD and are within symbols where not visible. Black bars indicate statistically significant differences, where p-values are *<0.05 and **< 0.01.

run-out, confirmed by comparison to a control single chromosome control strain, dnaA(Ts) (**S7 Fig**). *dnaB:R164A* has an increase in DNA content density, with major peaks between 2 and 4, and 4 and 8 chromosomes, respectively, indicating an odd number of chromosomes. *dnaB:K180A* has a major chromosome peak of 4, with a secondary peak at 8 but with diffuse signal in between. *dnaB:R74A* has major chromosome peaks just to the right of the 4 and 8 integer markers and aligning with major peaks in *dnaB:R164A*, indicating an increase in odd

numbers of chromosomes despite maintaining normal cell size (**Fig 2C**). It is possible that the major peaks indicating odd number chromosomes for *dnaB*:*R74A* and *dnaB*:*R164A* are in fact even numbered and do not align perfectly with the other strains. *dnaB*:*R328/9A* has the greatest and broadest increase in chromosome density, with a single major peak past 8 chromosome and even more signal to the right.

While the primary peaks represent integer numbers of chromosomes, significant broadening of peaks and loss of definition are seen for several of the strains, especially with the highest DNA content. This broadening at the 4 and 8 integer marks for *dnaB*:*K180A* and the indistinct right-shouldered single peak seen for *dnaB*:*R3288/9A* represent a subpopulation of cells that were unable to complete replication during the substantial 4-hour 'run-out' period. As *E. coli* grown at 32˚C can take up to 60 minutes to replicate their entire genome [39], these partially replicated chromosomes are indicative of significant blocks to DNA replication, termination, and/or chromosome segregation. The intermediate odd numbered peaks observed for *dnaB*:*R74A* and *dnaB*:*R164A* further suggest issues of replication stress, possibly from an accumulation of unresolvable replication intermediates, asynchronous initiation, delays in cell division, or defects in segregation [40,41].

To further elucidate root causes of increased and asynchronous chromosome numbers, we next performed a qPCR analysis, comparing the relative abundance of the origin (*ori*) and termination (*ter*) regions of the chromosome (**Fig 4B**). In *E. coli*, exponentially growing cells

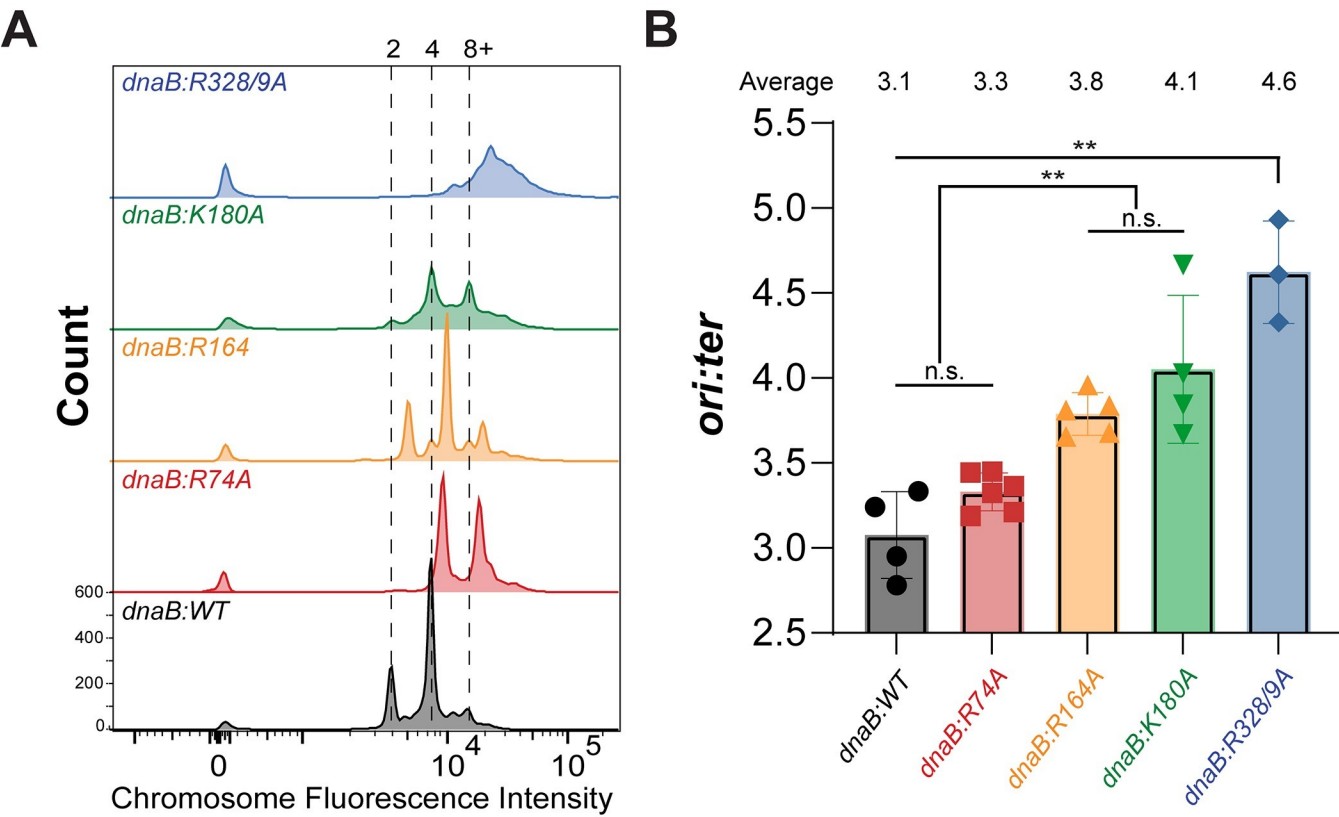

**Fig 4. *dnaB*:*muts* show changes in chromosome complexity.** (**A**) Chromosome density was measured by flow cytometry (FACS) in exponential growth phase rifampicin 'run-out' cultures stained with Sytox Green (n = 10,000 events). Chromosome integers are indicated at the top of the graph. (**B**) The *ori*:*ter* fold-difference measured by qPCR was calculated using the $2^{-\Delta\Delta Ct}$ method. Individual data is presented with symbols and the average values provided at the top of the plot. Black bars indicate statistically significant differences. Error bars represent ± CV; Black bars indicate statistically significant differences where p-values are ** < 0.01. n.s. is not significant.

generally have an *ori*:*ter* ratio ~3 [42], similar to our control strain (3.1 ± 0.4). Notably, *dnaB*: *R328/9A*, which had the highest and least distinct FACS chromosome signal also had an *ori*:*ter* ratio of 4.6 ± 0.3, significantly more than any of the other strains. Similarly, *dnaB*:*K180A* had a qPCR *ori*:*ter* ratio of 4.1 ± 0.3, and *dnaB*:*R164A* had an *ori*:*ter* ratio of 3.8 ± 0.2, both significantly higher than that of *dnaB*:*WT*. *dnaB*:*R74A* had an *ori*:*ter* ratio of 3.3 ± 0.2, which was not significantly different than that of the control. 3 of the 4 *dnaB*:*mut* strains had increased *ori*:*ter* ratios, likely from delays in elongation, termination, or segregation and consistent with the increased chromosome complexities visualized by FACS (**Fig 4A**).

## DnaB (R328/9A) is loaded onto ssDNA less efficiently

Previously, it was shown that a constricted DnaB is loaded less efficiently than WT and is generally unable to support rolling circle *in vitro* DNA synthesis [27]. To test whether there were any perturbations in DnaC dependent loading of these mutant DnaBs onto M13 ssDNA to indirectly explain differences in *ori*:*ter* ratios, we utilized a size exclusion chromatography loading assay [28]. Only those DnaB hexamers that are stably loaded on to M13 DNA would shift their elution profile toward the void volume (**Figs 5A and S8**). DnaB loading was monitored directly by identifying shifted fractions containing loaded DnaB (and DnaC) by Coomassie stain and M13 DNA by SYBR Gold staining correlating to the chromatogram ($A_{280}$). Unloaded DnaB/DnaC elute over a range of volumes consistent with several oligomeric conformations. The integrated and isolated peak regions were then used to calculate the % DNA loading efficiency using ATP as a convenient internal standard to account for any variability in the chromatographic runs according to **Eq 5** (**Fig 5B**). Although there were some moderate changes in DnaB loading efficiency across the mutants, DnaB (R328/9A) had a significantly reduced loading efficiency compared to WT, 18 ± 3% and 32 ± 4%, respectively. Loading of DnaB (K180A), 43 ± 5%, is increased slightly over that of WT, but it is not significant at the 95% confidence level; however, it is significantly different from that of DnaB (R328/9A). Neither DnaB (R74A) or (R164A) loading was significantly different from WT at 21 ± 5% and 36 ± 3%, respectively. Therefore, increased *in vitro* DnaC dependent loading of DnaB (K180A) may partially explain its increased *ori*:*ter* ratio *in vivo*, however, this is not the case for DnaB (R328/9A) as it has an impaired ability to load onto ssDNA.

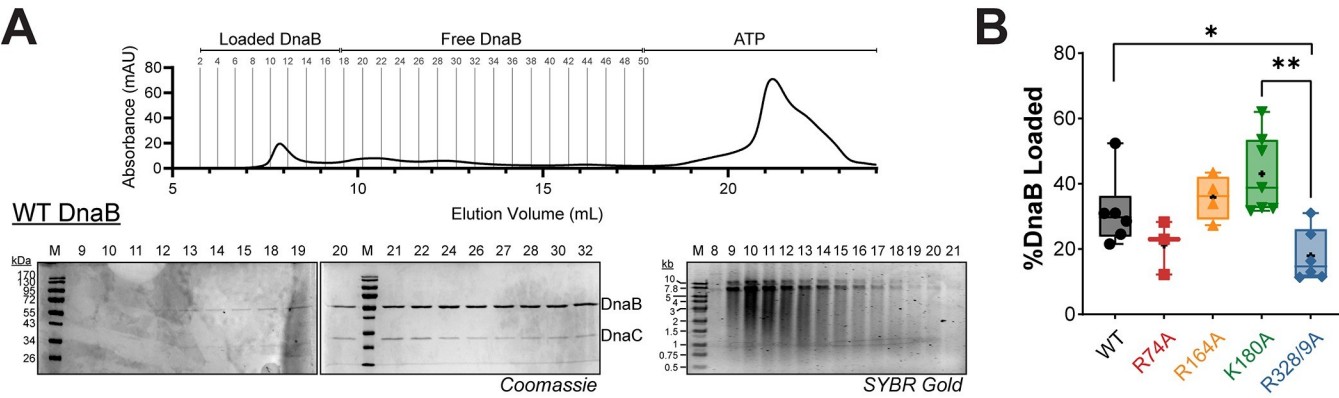

**Fig 5. Size exclusion chromatography DnaB loading assay.** (**A**) Purified DnaB enzymes were preincubated with purified DnaC, M13, and ATP before injecting onto a pre-equilibrated S200 10/30 size exclusion column, according to the Materials and Methods. Example chromatogram and associated SDS-PAGE (Coomassie) and agarose (SYBR-Gold) gels used to monitor loaded DnaB and Free DnaB areas. (**B**) %DnaB loading efficiency was calculated according to **Eq 5** and all experimental runs plotted (white circles) as box and whiskers for all the DnaBs, where the median is indicated as a while line. Black bars indicate statistically significant differences where p-values are ** < 0.01 and * < 0.05.

### *dnaB:K180A* and *dnaB:R328/9A* have increased levels of mutagenesis

The poor competitive nature of our helicase mutants coupled with signs of genomic stress naturally led us to investigate their effect on genomic stability. Whole genome sequencing (WGS) was performed for all *dnaB:mut* strains and compared to the parental strain to identify any potential suppressors and characterize mutations. Two independent WGS runs were utilized for higher confidence single nucleotide polymorphism (SNP) identifications. No obvious suppressor mutations in DNA replication or repair genes were observed (**S1 Table**); instead, the SNPs appeared to occur randomly across the genome (**Fig 6A**) with a higher frequency of transition mutations (average of 0.87 across all strains) (**Figs 6B and S9**) consistent with a general mutator phenotype. The parental strain is MutS⁻, deficient for mismatch repair (MMR), and so, the mutational spectrum represents all misincorporations that arise during replication. Previously, it was reported that MMR deficient *E. coli* strains have a high transition preference [43] that is similar but enhanced in our *dnaB:mut* strains.

We measured the mutation frequency of our *dnaB* strains by testing for rifampicin resistance (**Fig 6**). Briefly, cells were grown in LB media and then exposed to lethal levels of rifampicin [37,44,45]. The number of resistant colonies that arose determined the mutational frequencies. *dnaB:R328/9A* has the highest frequency at $15.0 \pm 3.8$ mutation events per $10^6$ cells, followed by *dnaB:R164A* with $6.6 \pm 0.9$ and *dnaB:K180A* with $4.4 \pm 0.4$ events per $10^6$, and all were statistically significant compared to the control strain at $0.6 \pm 0.7$. The slightly lower mutagenesis for *dnaB:K180A* could be due in part to its slower growth (**Fig 2A–2B**) and

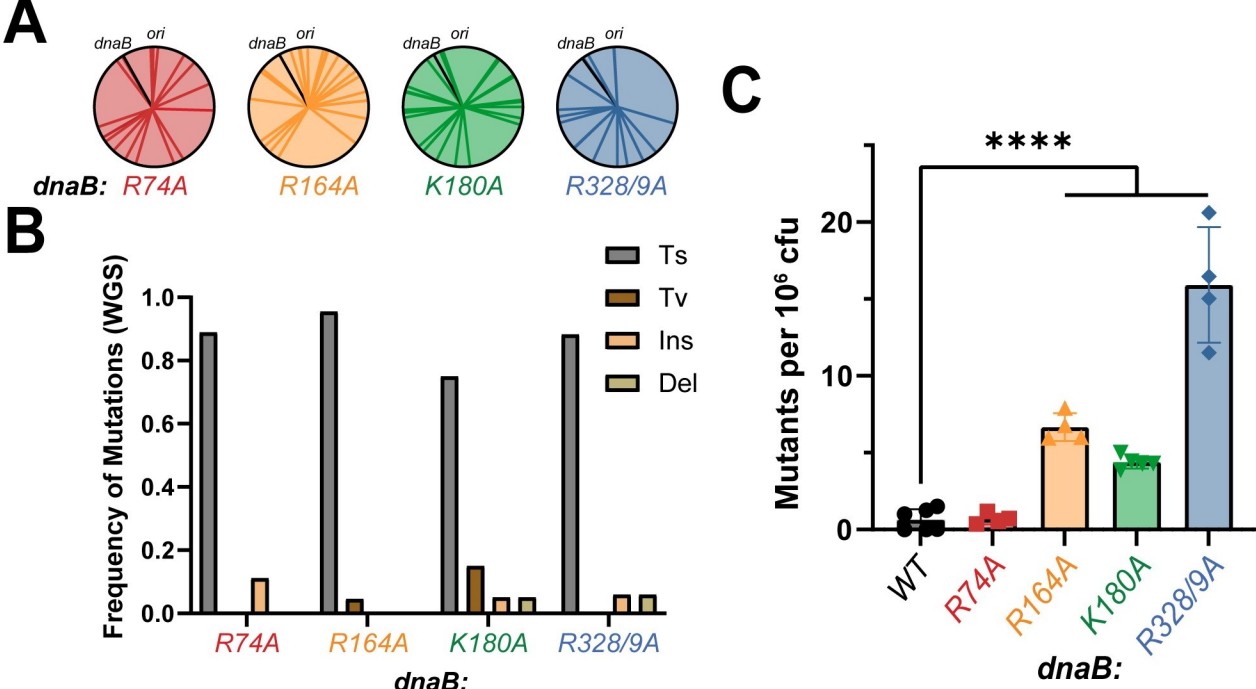

**Fig 6. *dnaB:muts* have increased frequency of mutations. (A)** A visual representation of the positional location of mutations detected by two independent whole genome sequencing (WGS) runs mapped on to the circular *E. coli* chromosome for each *dnaB:mut*. **(B)** Quantification of the frequency of mutation types. Ts—transitions, Tv–transversions, Ins–insertions, Del–deletions. Further characterization of Ts and Tv are plotted in **S10 Fig** and individual mutations are provided in **S1 Table. (C)** The mutagenicity of each strain was monitored using a rifampicin resistant assay to quantify number of rif^R colonies after 24-hour growth in LB. Average number of resistant colonies per 1 million CFU's are shown (**Eq 6**). Data is from at least two trials of three technical replicates each (n ≥ 6). Error bars represent ± SD; Black bars indicate statistically significant differences, where p-values are ****< 0.0001.

competitiveness (**Fig 3**) compared to *dnaB*:*R164A*. Although *dnaB*:*R74A* demonstrated poor fitness (**Fig 2**) and increased chromosome number (**Fig 4**), this strain did not show a mutational frequency that was significantly different to that of the parental strain in this assay, even though several mutations were detected by WGS (**S1 Table**). Notably, *dnaB*:*R328/9A* had the largest increase in cell length (**Fig 2C**) and the greatest chromosome complexity (**Fig 4B**) correlating with the highest mutational frequency out of all the mutant strains.

## SOS is not significantly induced in any of the *dnaB*:*muts*

To determine if the increases in mutagenesis and chromosome complexity correlated with an abundance of mutagenic repair (especially for *dnaB*:*K180A* and *dnaB*:*R328/9A*), we investigated whether the SOS response was activated in *dnaB* mutant strains. The bacterial SOS response is induced when excess ssDNA intermediates are available for RecA polymerization, a common event under DNA damaging conditions, leading to increased expression of mutagenic repair proteins [46,47]. We measured SOS induction by transforming our strains with a plasmid containing SuperGlo GFP under a RecN SOS-regulated promoter as performed similarly [48].

Interestingly, none of the *dnaB*:mut strains showed an induction of SOS response when grown exponentially in LB with no exogenous treatment (**Fig 7A**). It was not until low levels (0.001 µg/mL) of the crosslinking agent, mitomycin-C (MMC) was added to the media that *dnaB*:R328/9A started to show a significant induction of SOS over that of the parental strain (**Fig 7B and 7C**). The specific fluorescence (measured by **Eq 7**) was fitted with **Eq 2**, and the maximum SOS response was quantified for each strain. With low dose MMC, *dnaB*:*WT* had a peak specific fluorescence of $112 \pm 6$; *dnaB*:*R74A* had a peak specific fluorescence of $67 \pm 3$; *dnaB*:*R164A* had a peak specific fluorescence of $91 \pm 4$; and *dnaB*:*K180A* had a peak specific fluorescence of $69 \pm 4$. Unlike all other mutants, *dnaB*:*R328/9A* had significantly higher SOS induction (when MMC was included) compared to the parental strain, with a peak specific fluorescence of $250 \pm 46$, 2.5-fold greater than the parental strain.

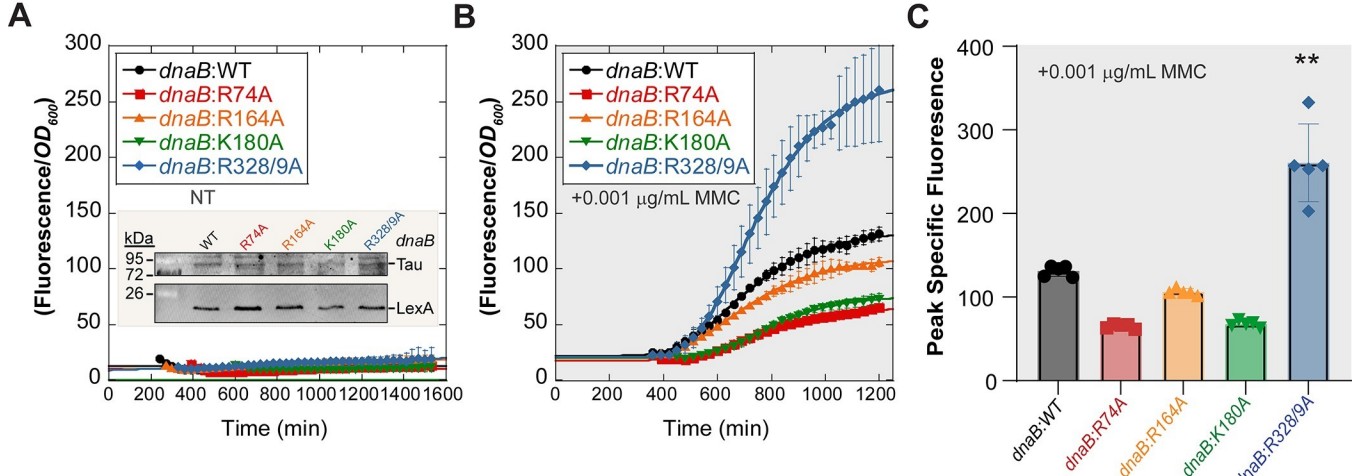

**Fig 7. The SOS response is not significantly induced in the *dnaB*:*mut* strains.** Specific fluorescence for *dnaB* strains grown (**A**) in LB with no exogenous treatment (NT) or (**B**) with low dose mitomycin C (0.001 µg/mL), each containing a plasmid expressing GFP under a *recN* promoter to monitor SOS induction. An inset in (**A**) shows a western blot of stable LexA expression in all *dnaB*:*muts* in NT conditions. Specific fluorescence is defined as the measured fluorescence divided by the optical density ($OD_{600}$) to control for growth. Due to the error inherent in dividing very small numbers, specific fluorescence is not shown during lag phase. (**C**) The maximum fluorescence from **B** is plotted for comparison between strains (**Eq 7**). Error bars represent $\pm$ SD and are within symbols where not visible. Statistically significant differences are indicated where p-values are **< 0.01.

### *dnaB:mut* strains have more ssDNA gaps and show a spectrum of lethality in *ΔrecA* strains

To specifically observe and compare differences in DNA damage, we performed a bacterial TUNEL assay to fluorescently label ssDNA and dsDNA DNA breaks. Exponential growth cultures were fixed and permeabilized before labeling DNA ends with BrdU and staining the nucleoid with DAPI. Stained cells were imaged, and representative microscopy images are shown (**Fig 8A**). Interestingly, there was a marked visible increase in BrdU foci for all mutant strains. As a positive control, we exposed an exponential growth culture of *dnaB:WT* to moderate doses of MMC [49], for 45 minutes before harvesting, which also showed significant increases in BrdU fluorescence. We also treated cells with hydroxyurea (HU) as a negative control and saw minimal TUNEL staining, confirming that our staining procedure did not cause aberrant breaks. Intriguingly, while the other three mutants show widespread and sporadic DNA breaks, the foci in *dnaB:R74A* were very bright and consistently located near the poles of the cell.

FACS analysis was performed to quantify BrdU abundance for all strains and showed a marked increase for all the *dnaB* mutants (**Fig 8B**). BrdU positive cells were gated at $10^3$, with *dnaB:WT* having 39.4% of cells with BrdU signal and the positive control (*dnaB:WT* + MMC) having 88.2% of cells with BrdU signal. *dnaB:R328/9A* had 77.1% of cells with ss and dsDNA breaks, followed by *dnaB:R164A* with 71.9%, and *dnaB:K180A* with 61.6%. *dnaB:R74A* had the highest number of cells with ss and dsDNA breaks, with 91.4% of the population containing BrdU signal, consistent with strong fluorescent signals at the poles in the vast majority of cells (**Fig 8A**). Noticeable tailing relative to the parental strain represents populations of cells that have increased DNA and DNA damage (as seen for *dnaB:R74A*) (**S10 Fig**).

## Discussion

In this work, we determined that targeted external DnaB surface mutations, previously shown to limit interaction with the excluded strand *in vitro* also stabilize a constricted hexamer

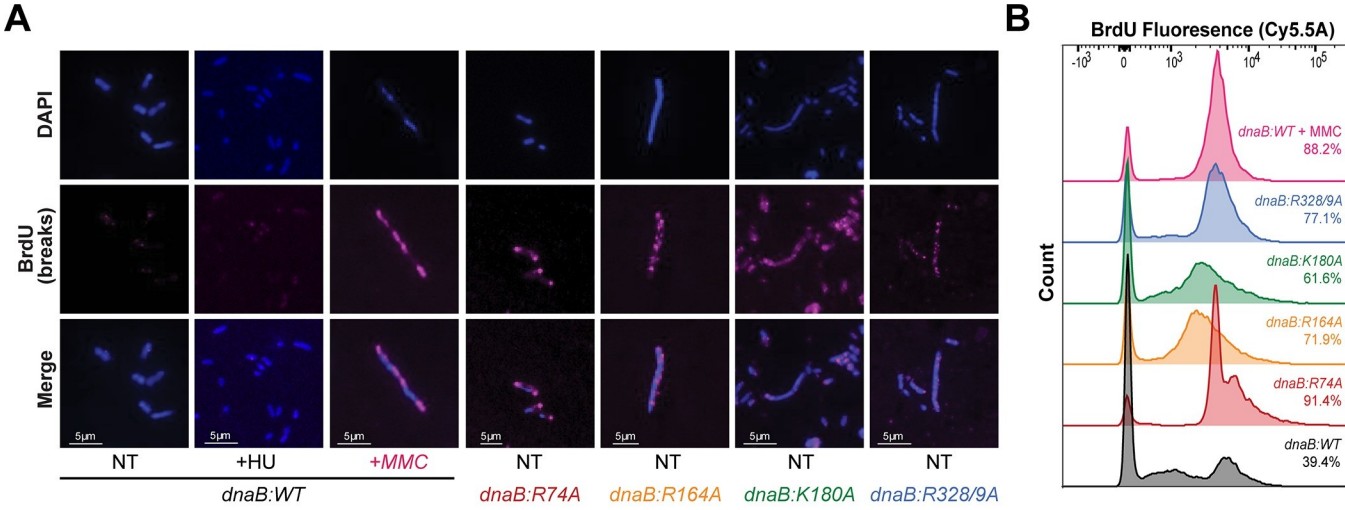

**Fig 8. *dnaB:mut* have increased DNA damage under normal conditions. (A)** Exponential growth cells were probed for DNA breaks by a TUNEL assay. Blue (DAPI) represents DNA staining, and pink (BrdU) represents tagged DNA breaks. NT–nontreated; +HU–treated with 10 mM HU; +MMC–treated with 0.01 μg/mL MMC. Images shown are representative of the population observed. The fraction of BrdU labeled cells was **(B)** measured and quantified by flow cytometry (FACS), utilizing the same exponential growth cultures as the microscopy images. BrdU-negative and positive populations were gated at $10^2$ and the percentages are indicated on the plot; data is from n = 10,000 events.

conformation. These mutations, when edited into the bacterial genome, cause widespread genomic and replication stress, alter cellular morphology, and limit cellular fitness. Specifically, *dnaB* edited strains present with increased DNA damage and mutagenic repair and display signs of replication progression and/or termination defects, *even* in the absence of any external genotoxic stress. Although there may be other unknown explanations, the spectrum of deficiencies in the *dnaB:mut* strains correlate with the degree of constriction in the DnaB hexamer, where the most constricted mutants, *dnaB:K180A* and *dnaB:R328/9A* have the most severe deficiencies. These findings highlight the functional importance of dynamic helicase regulation for faithful and efficient replication.

Based on current research, it is suggested that the helicase conformation plays a role in DNA loading, unwinding efficiency, protein associations at the fork, and helicase-polymerase coupling [14,27]. In the event of polymerase pausing or stalling, helicase activity should slow, likely changing conformation to engage with the excluded strand and minimize forward progression [28,50,51]. DnaG, which favors the dilated state of DnaB, has been shown to limit replisome progression and generate pausing events [52], consistent with a model where the helicase constricts for efficient unwinding and dilates to slow unwinding and allow for more efficient priming, however the mechanism and impact of helicase-primase interaction is still being investigated [26,53]. It is unclear whether during observed pausing events, the helicase and polymerase physically decouple, but it is apparent that ssDNA continues to be generated with an approximate 10-fold reduction in rate [25]. Our electrostatic SEW mutants stabilize a spectrum of constricted states, disrupt interactions with the excluded strand [4] and are poised for rapid unwinding, suggesting that their helicase activity should be continuous and fast and independent of Pol activity, possibly to the detriment of cell proliferation and organismal fitness. However, while one function of helicase regulation is clearly to limit ssDNA production, a lack of increased SOS response indicates that the function of helicase regulation goes beyond limiting unwinding and impacts other aspects of replisome activities.

While all of the selected DnaB residues were predicted to be surface exposed based on homology modeling with a *Geobacillus stearothermophilus* DnaB crystal structure, the two more prominent SEW sites, K180 and R328/9, appear to be partially buried in a more recent crystal structure [28] of a constricted *E. coli* DnaB (**S1 Fig**). When mutated, DnaB (K180A or R328/9A) stabilize a fully constricted state. DnaB unwinding activity is also greatly stimulated by the addition of tau [14], which stabilizes a constricted conformation and would be present in an active replisome serving to coordinate polymerase and helicase activity. We therefore suggest that key SEW residues are concealed when DnaB is constricted to allow fast and efficient replisome progression but become exposed in the dilated state to limit helicase activity (**Fig 9**). Interestingly, a constricted state of DnaB has a higher affinity with the CLC and may possibly explain the rapid exchange/recruitment of the Pol III core (Pol III*) seen in *in vitro* assays [54], single-molecule [55], and *in vivo* imaging experiments [56]. Additionally, we note that both K180 and R328/9 are located in the C-terminal domain (CTD) of the enzyme; R164 is located in the linker helix; and R74 is located in the N-terminal domain (NTD) (**S1 Fig**). While the N-terminal collar significantly contributes to the helicase conformation [14,27], it is clear that each domain of the helicase plays a role in the conformational dynamics and interactions within the replisome.

Although DnaB(R74A) and (R164A) moderately stabilize the constricted state, the phenotypic results with these edited strains were overall less deleterious than either K180A or R328/9A. The ability to dilate, even infrequently, likely contributes to the more regular size, stable *ori:ter* ratio and chromosome density, and lower mutagenesis frequency of the *dnaB:R74A* strain. However, while *dnaB:R164A* also had low mutation frequency, it displayed genomic stress markers and generally poor fitness despite having a near identical dilation frequency.

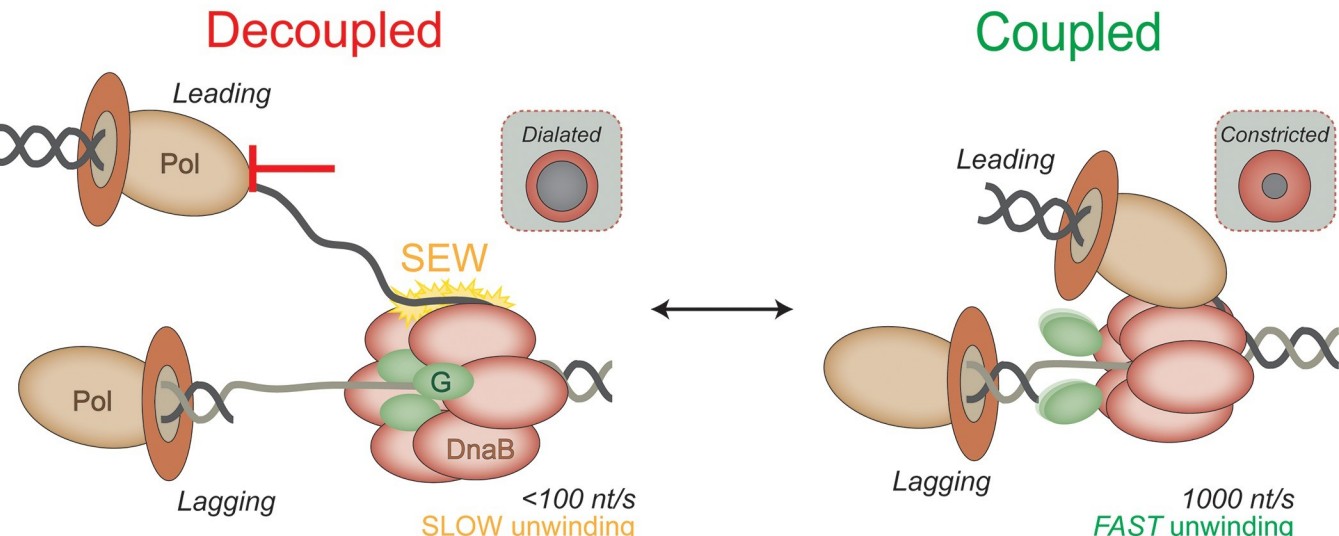

**Fig 9. Model for helicase-polymerase decoupling leading to genomic instability.** DnaB unwinding is rapid when coupled tightly within the replisome. Upon encountering various obstacles to DNA synthesis, DnaB adopts a dilated conformation and interacts with the excess excluded strand to slow the unwinding rate and limit production of ssDNA, allowing the polymerase (Pol) to couple with DnaB. DnaG (green) is more tightly associated with a dilated DnaB. DnaB mutants resist conformational switching to the dilated state and maintain accelerated unwinding, increasing the amount of labile ssDNA produced causing genome instability.

This highlights the importance of dilation and excluded strand engagement for normal, damage-resistant replication, while also making it evident that moderate dilation alone is not enough to restore normal or even mostly normal function.

Interestingly, mutation of *dnaB*:*R74A* in the NTD of the protein had the least impact on cellular growth and morphology, with low mutagenesis and SOS, and was the only strain to maintain normal *ori*:*ter* ratios. Despite this, it had the highest fraction of cells with BrdU foci, localized primarily at the poles, and a large increase in chromosome number. This polar TUNEL localization was consistent across images and indicates that DNA breaks may be localized to the *ori* and/or *ter* regions of the chromosome, which migrate to the ends of the cell during chromosome segregation [57–59]. Although DnaC loading of DnaB (R74A) was not significantly affected, these highly localized breaks suggest specific issues with replication initiation and/or termination, which may instead be the result of altered interactions with DnaA and/or Tus [60,61]. Despite this, *dnaB*:*R74A* maintained normal growth rates, cell size, and *ori*:*ter* ratio, but decreased fitness compared to the parental strain.

While most of the behavior of *dnaB*:*R164A* is similar to the other mutants, it was the only strain to have elongated cells even in stationary phase. The R164 residue is in the linker helix domain of DnaB, which is responsible for coordinating the hexamerization of the helicase from monomer into its active form [62] and is responsible for interaction with the primase DnaG [63]. The location of the mutation may contribute to the observed phenotype of this strain. In addition to slow growth, elongated cells, and poor competitiveness which was consistent for most strains, *dnaB*:*R164A* exhibited strong asynchronous initiation, as indicated by chromosome numbers other than $2^n$ (but without peak broadening) and a unique cell population with high chromosome complexity but smaller cell size. *dnaB*:*R164A* also had widespread ss and dsDNA breaks, similar to *dnaB*:*K180A* and *dnaB*:*R328/9A*.

The genomic consequences that result from altering the equilibrium of DnaB dilation are severe, especially when enforcing a static fully constricted state as with *dnaB*:*K180A* and *dnaB*:*R328/9A*. All of our *dnaB*:*mut* strains can be classified as mutator strains based on the number

and distribution of genomic mutations as well as the increased frequencies of rifampicin resistance. This, combined with significant detected 3' ends, increased genomic complexities, and increased cell sizes even in the absence of damage suggests that recombination processes may be hyperactive as a means to compensate for replication fork deficiencies. Interestingly though, in the absence of damage, the measured SOS response is not significant and only becomes apparent in *dnaB*:*R328/9A* when further stressed with low doses of MMC.

The failure to initiate a significant SOS response even under conditions of overwhelming cellular and genomic stress is somewhat troubling to reconcile. In light of this observation, the interpretation of the TUNEL results may be more consistent with single-strand gaps over that of double strand breaks. In fact, looking more carefully at the TUNEL staining of the nucleoid, the TUNEL spots appear to transverse randomly along the length of cells. Therefore, these single-strand gaps may be filled with a more error prone DNA polymerase, such as Pol IV, which is already at high enough levels within the cell even in the absence of SOS induction [64], explaining the increased mutagenesis detected through WGS and rifampicin resistance. The other major error prone DNA polymerase in *E. coli*, Pol V, is proteolytically activated by significant RecA filamentation to cleave the UmuD subunit to create the active heterotrimeric UmuD'$_2$C complex [65,66], but as SOS is not significantly induced in our strains, Pol V is less likely to be contributing to the observed mutator phenotype. Finally, if single-strand gaps become prevalent in these *dnaB*:*mut* strains, then there is a question on whether they exist primarily on the leading or the lagging strands. Gaps on the lagging strand can easily be filled in with Pol IV as suggested, however, leading strand gaps required some additional priming and restart and may require RecFOR or a specific subset of the PriA/B/C pathways depending on the size of the gap or conformation of the stalled fork [67,68]. The genomic consequences to this helicase conformation induced decoupling will require further investigation.

Helicase regulation has been shown vital to maintaining a stable DNA duplex in the event of discontinuous replication. Recent work suggests that rather than a smooth and continuous process, replisome progression is naturally stochastic [22] with frequent pausing and polymerase exchange [56], indicating that decoupling is a natural component of the replication process (**Fig 9**). The effect of these *dnaB* mutations on strain growth, fitness, and chromosome complexity, even in the absence of environmental stress or exogenous damage, support the notion of frequent replisome pausing and helicase-polymerase decoupling, which if not regulated could lead to ssDNA buildup, lack of replisome coordination, and reduction in fidelity. This agrees with previous observations that proposed helicase progression is a vital fail-safe mechanism to maintain helicase-polymerase coordination during polymerase pausing and allow time for TLS or repair in the event of DNA damage [26]. More binary rapid unwinding kinetics by a constricted helicase cannot adequately respond to stochastic DNA polymerase progression, likely contributing to decoupled synthesis and unwinding activity, production of labile ssDNA, less frequent priming, and challenging DNA structures migrating behind to limit synthesis. These effects compound with initial replication challenges, encouraging slow, inefficient replisome progression and resulting in poor performance as seen with our SEW mutants.

The spectrum of effects of these *dnaB* mutants on faithful and efficient replisome activity is complex and confirms the importance of helicase dynamics within the replisome as a whole. Disruption of this process, however minor, will amplify with replication and environmental stress *in vivo*. Replisome cohesion is a critical aspect of the protein complex, as it harnesses many mobile elements that work in stochastic harmony. Helicase conformation and SEW are jointly responsible for regulating not only helicase unwinding activity, but also controlling aspects of replisome coordination (loading, priming, unwinding) and progression (DNA unwinding speed) in the presence of various genomic obstacles. Our data suggests that both

speed and helicase-protein interactions contribute to the deleterious effects seen for DnaB malfunction *in vivo*, and further studies are needed to elucidate the specific role and impact of the helicase-polymerase tether, the CLC, and whether decoupling occurs at sites of damage.

## Materials and methods

### Purification of *E. coli* DnaB and mutants

Wild-type *E. coli* DnaB and mutants (R74A, R164A, K180A, R328/329A) were independently expressed in C41 strain (Lucigen, Middelton, WI) from pET11b-derived plasmids as previously described [4]. Briefly, IPTG (1 mM) induced DnaB was pelleted and resuspended in the lysis buffer (20 mM Tris [pH 7.5], 500 mM NaCl, 10% glycerol, 10 mM $MgCl_2$, 1 mM PMSF) and lysed using lysozyme and sonication. Ammonium sulfate (0.17 g/ml) was added to the resultant supernatant, pelleted, and then resuspended in Buffer A (20 mM Tris [pH 7.5], 10 mM $MgCl_2$, 0.1 M NaCl, 10% glycerol, 0.01 mM ATP, 1 mM BME) and applied to HiTrap MonoQ column (Cytiva, Marlborough, MA) and eluted with a stepwise gradient of buffer A supplemented with 0.75 M NaCl. The fractions were combined and applied to Superdex S-200 26/600 gel filtration column (Cytiva, Marlborough, MA) with Buffer B (20 mM Tris [pH 7.5], 5 mM $MgCl_2$, 0.8 M NaCl, 10% glycerol, 0.1 mM ATP, 1 mM BME). Combined peak fractions were concentrated and dialyzed against storage buffer (20 mM Tris [pH 8.5], 500 mM NaCl, 5 mM DTT, 50% glycerol).

### DNA translocation assay

The translocation assay was performed as described [14] in a 96-well plate (Corning) using a Varioskan Lux Microplate Reader (Thermo Scientific, Waltham, MA). Substrates (**S2 Table**) were purchased from Integrated DNA Technologies (IDT, Coralville, IA). Annealing reactions were performed in a solution of 10 mM Tris-HCL, 1 mM EDTA, and 100 mM NaCl (pH 7.5) using a thermocycler following the protocol 95˚C for 6 min, then decreases 1˚C/min with a final hold at 25˚C. Translocation reactions contained 20 nM substrate and 40 nM DnaB hexamer in reaction buffer (20 mM HEPES-KOH [pH 7.5], 5 mM magnesium acetate, 50 mM potassium glutamate, 5% glycerol, 0.2 mg/mL BSA, 4 mM DTT,), and initiated with 1 mM ATP and 200 nM trap (unlabeled DNA165) (**S2 Table**). Data was fit to the following equation for a single exponential:

$$\%DNA\ unwound = A_0 + Ae^{-kt} \tag{1}$$

where $A_0$ is the lower asymptote, $A$ is the amplitude, $k$ is rate and $t$ is time.

### CRISPR-Cas9 genomic editing

Using a dual-plasmid system designed for *E. coli* engineering [69], we created four distinct strains that carry precise mutations in the gene (*dnaB*) that codes for the replicative helicase DnaB (**S3 Fig** and **S3 Table**). The parental strain, HME63, is a derivative of *E. coli* W3110 with Δ*mutS* for suppression of mismatch repair and optimized for recombineering by the expression of λ-red genes (*gam*, *exo*, and *bet*) under control of a temperature sensitive (ts) repressor [70,71]. All mutant *dnaB* strains are derivatives of HME63, created using dual plasmid CRISPR-Cas9 system to generate a point mutation in the DnaB helicase enzyme by chromosomal DNA modification [69,72–74]. Briefly, 30 base regions of the *dnaB* gene (**S2 Table**) targeting the mutation site were inserted into the pCRISPR plasmid (Addgene: 42875) as a guide RNA (gRNA) and electroporated into HME63 already containing pCas9 (Addgene: 42876). The *dnaB* gRNA was designed to be centered on the desired mutation site and at the closest

adjacent PAM sequence (5'-NGG). 1 µM of a synthetic 60 base ssDNA oligonucleotide template for homologous recombination (HR) containing the desired mutation on the lagging strand was simultaneously electroporated with 100 ng pCRISPR to edit the bacterial genome (S2 Table). HME63 electroporated with pCRISPR and pCas9 but without the editing oligonucleotide were used to determine background levels. Precise genome editing was monitored using PCR and screening for a novel engineered restriction site (S3 Fig) before confirmation by DNA sequencing of the locus (UT Austin, Genome Sequencing and Analysis Facility). All confirmed strains were then outgrown in the absence of Kan/Chl to remove CRISPR-Cas9 editing plasmids, reconfirmed for the *dnaB* mutation by PCR and restriction digest, and frozen as working stocks. All subsequent cultures were grown in LB (10 g typtone, 10 g NaCl, 5 g yeast extract per L, pH 7.0) supplemented with 100 µg/mL ampicillin, and all incubation steps were performed at 32˚C, unless otherwise stated. All strains are listed in S3 Table.

## Growth curves

Growth curves were recorded by diluting overnight clonal cultures to $OD_{600} \sim 0.01$ in LB and aliquoting 200 µL into a black-walled clear-bottomed 96-well plate (Corning). The cultures were incubated at 32˚C with aeration at 225 RPM and the $OD_{600}$ was recorded at 30-minute intervals using a Varioskan Lux microplate reader equipped with SkanIt 5.0 software (Thermo Scientific, Waltham, MA). Data was processed using KaleidaGraph (Synergy Software, v.4.5.3) and fit to a modified 4-parameter Gompertz growth model [75] according to the following equation:

$$w(t) = B + A^{-e^{\left(\frac{k_g \times 2.7182}{A} \times (T_{lag} - t) + 1\right)}} \tag{2}$$

where *w(t)* is the density as a function of time, *B* is the lower asymptote, *A* is the upper asymptote, $k_g$ is the growth rate coefficient, $T_{lag}$ is the lag time of the culture, and *t* is time. The absolute growth rate ($k_z$) is calculated by

$$k_z = \frac{k_g \times 2.7182}{A}. \tag{3}$$

## Microscopy

All microscopy images were obtained using an Olympus Brightfield Microscope IX-81 (Olympus Corp., Center Valley, PA) using a 60x objective with oil immersion. For stationary phase cells, 1 mL of each culture was grown overnight in LB/Amp at 32˚C, pelleted, washed in PBS, and then fixed with 70% ethanol. 2 µL of the fixed sample was spotted onto a microscope slide and allowed to dry. DAPI (Thermo Fisher, Waltham, MA) was added to mounting media (2.5% DABCO, 90% glycerol, 7.5% PBS) to create a dual staining and mounting solution. 3 µL of this solution was added to cover the fixed cells, then immediately topped with a coverslip and sealed with clear polish. Slides were stored at 4˚C in the dark overnight prior to imaging. For DAPI-only stained exponential growth cells, overnight cultures were diluted 1:1000 in LB and grown with aeration until OD ~ 0.2–0.4 before following the same procedure described for stationary phase cells. Filamentation was quantified by blinded counting and measuring $\geq 200$ cells for each condition using Image J [76]. This data was analyzed by excel and plotted using Prism 9.1 (GraphPad, San Diego CA).

## Red-white strain competition assays

The strain fitness was determined using a red-white assay as described [31]. Construction of the control strain was done using P1 phage transduction. The phage was harvested from the

cell line EAW214, a derivative of MG1655 engineered to contain a mutant FRT-KanR-wtFRT cassette in place of the araBAD promotor [31]. Through transduction, the neutral *ΔaraBAD* mutation was transferred to the parental strain (HME63) and designated MSB1 (*HME63:Δara-BAD*). Overnight cultures were mixed 1:1 and then diluted $10^{-8}$ in sterile PBS and plated onto tetrazolium arabinose (TA) indicator plates, containing 0.2 mg/mL triphenyl tetrazolium chloride (Sigma, St. Louis, MO) and 1% arabinose (Oakwood Products, Estill, SC), and then grown at 32˚C for 24–36 hours before colonies were counted and sorted by color. MSB1 will have red color when grown on TA plates and allows for easy colorimetric differentiation between the *dnaB* mutants (ara$^+$) strains. Mixed cultures were sampled and diluted 1:100 in fresh media every 24 hours until a strong divergence in population was seen or 72 hours was reached. Relative selection rate was calculated according to the following equation:

$$selection\ rate/day = (r) = \ln\frac{A1}{A0} - ln\frac{B1}{B0} \qquad (4)$$

where $A_0$ and $B_0$ are the CFU fraction of strains A and B at time 0, and $A_1$ and $B_1$ are the CFU fraction of strains A and B after 24 hours.

## Flow cytometry

To quantify cell size and chromosome number, overnight cultures were diluted and grown until mid-exponential phase (OD$_{600}$ ~ 0.3) before treatment with 150 mg/mL rifampicin and 10 mg/mL cephalexin (TCI America, Portland, OR). Samples were then incubated with shaking for 4 hours to allow for completion of chromosome synthesis (or 'run out'). Samples were pelleted, washed with cold TE buffer, and then fixed in 70% ethanol. Fixed cells were pelleted, washed in filtered PBS, and then resuspended in 500–1000 μL TBS containing 1.5 μM Sytox Green (Invitrogen, Carlsbad, CA) and 50 μg/mL RNaseA for 30 minutes at 4˚C in the dark. Stained samples were pelleted and then resuspended in sterile PBS before analysis using a FACSverse (BD Biosciences, San Jose, CA). The parental strain was used as a chromosome reference; run-out of normal actively replicating cells gives major peaks for 2 and 4 chromosomes used as reference for chromosomal number. A control experiment comparing the parental strain with CM742 containing a mutant of *dnaA* (*dnaA46(*Ts), (**S3 Table**) [77] to confirm chromosome numbers was performed (**S7 Fig**). To prepare, an overnight culture of CM742 was diluted and grown to mid-exponential phase (OD$_{600}$ ~ 0.3) at 30˚C, transferring to a 42˚C water bath for 5 minutes and then incubating at 42˚C for 2 hours to synchronize cells [78]. 5 mL of this synchronized culture was diluted with 5 mL of cold 4˚C media to rapidly return the cells to the permissible temperature. The cooled cells were then allowed to shake at 32˚C for 10 minutes to allow for one round of replication initiation, before re-heating with 10 mL of warm (55˚C) media and transferring to a 42˚C water bath for 5 minutes. The culture was then transferred to a 42˚C shaker for 170 minutes to allow for run-out of replication. Cells were harvested by pelleting and fixing in cold (-20˚C) 70% ethanol. Fixed cells were stored in the fridge until analysis. Data was plotted and presented using FloJo software (BD Biosciences).

Quantification of DNA breaks was performed using the TUNEL assay described above and quantified using FACS. Antibody-stained cultures were pelleted, resuspended in 1 mL sheath fluid (sterile 1x PBS) with 1.5 μM Sytox Green, and incubated in the dark for 30 minutes. Samples were diluted with an additional 1 mL sheath fluid before being analyzed by FACSverse. HME63 exposed to 0.01 μg/mL MMC prior to harvesting was used as a positive control; Sytox Green alone and fluorescently labeled BrdU alone were used as signal controls. Data was plotted and quantified using FloJo software.

## qPCR

To determine the relative chromosomal complexity for the mutant and parental strains, quantitative PCR was performed using a Quant-Studio 6 Flex Real-Time PCR instrument (Thermo Scientific, Waltham, MA). For sample preparation, overnight cultures were diluted 1:1000 in 3 mL LB, grown at 32°C until OD ~ 0.5–0.85, and then pelleted and fixed in 70% ethanol. Harvested cultures were kept at 4°C. Before use, fixed cells were washed and resuspended in 1 mL sterile water. The qPCR assay was performed using a PowerUp SYBR green master mix (Applied Biosciences, Beverly Hills, CA), 1 μL resuspended cells, and 0.5 μL each of 10 μM forward and reverse primers (**S2 Table**). The *ori:ter* qPCR ratio was calculated using the $2^{-\Delta\Delta Ct}$ method for comparative cycle threshold (Ct) analysis [42,79]. A fixed overnight sample of the parental strain, where the population would have an *ori:ter* ~ 1, was used for normalization in every run. Each sample had a minimum of five technical replicates in each cycling run, and the mean Ct value was used to calculate the *ori:ter* ratio.

## DnaB loading assay

Purified *E. coli* DnaB WT and mutants (3 μM hexamer) and *E. coli* DnaC (18 μM monomer) were mixed with 11 μg of M13 ssDNA (Guild Biosciences, Dublin, OH) to a final volume of 250 μL in the presence of 20 mM Tris pH 8.5, 200 mM NaCl, 5% glycerol, 5 mM MgCl$_2$, 5 mM ATP. Reactions were incubated for 10 min at 37 °C and then applied to a Superdex 200 10/300 size exclusion column equilibrated in the same buffer minus ATP using AKTA Pure 25L (Cytiva, Marlborough, MA). 250 μL fractions were collected and representative fractions were resolved on both 15% SDS-PAGE and 1% agarose gels. The gels were stained with either Coomassie or 1X SYBR Gold (Thermo Scientific, Waltham, MA), respectively, imaged using Gel Doc EZ gel documentation system (BioRad), and analyzed to determine the fractions containing both DnaB and ssDNA. The area (ml*mAu) under the chromatographic curve was integrated for the separate regions from the elution profile containing: 1) DnaB loaded onto M13 ssDNA (DNA$^L$), 2) unloaded combinations of DnaB and DnaC (DNA$^F$), and 3) ATP alone using Unicorn 7.1 (Cytiva, Marlborough, MA). The DnaB loading efficiency was calculated as a ratio of the loaded DnaB area over that of the total DnaB area normalized to the area of the ATP peak as an internal standard according to:

$$\%DnaB\ Loaded = \left( \frac{\frac{DnaB^L}{ATP} - \frac{M13}{ATP}}{\frac{DnaB^L}{ATP} + \frac{DnaB^F}{ATP}} \right) \times 100 \tag{5}$$

The A$_{280}$ signal contributed by M13 alone (in a separate chromatograph run without protein) was subtracted from peak 1 again using ATP as a normalized internal standard. The experimental data from at least three independent experiments were averaged, plotted as a box and whiskers, and analyzed for significant differences from WT DnaB using parametric t-tests in Prism 9.1 (GraphPad, San Diego CA).

## Whole genome sequencing

Cells were grown in LB media at 32°C from 1:1000 dilution to OD~0.2. Genomic DNA was prepared from 10 ml culture by the CTAB method [80] and sequencing libraries were prepared using the Nextera XT Sample Preparation Kit (Illumina) from 1 ng of genomic DNA. Paired-end sequencing was performed on an Illumina MiSeq sequencer using the re-sequencing workflow with a 2×75-cycle MiSeq Reagent Kit v3 (Illumina). Sequencing reads were mapped to the *E. coli* W3110 reference genome (NCBI RefSeq accession: NC_ 007779.1) using MiSeq integrated analysis software. All sequencing data generated in this study have been deposited

to the Sequence Read Archive (accession no. PRJNA773110). Mutation analysis was done by mapping sequencing reads (average 2 million reads per sample) to the *E. coli* W3110 reference genome and identifying the number and prevalence of variants using the rapid haploid variant calling program snippy v4.5.0 [81]. Variant calling thresholds were set to 10-fold coverage and 90% prevalence of variant among all reads in a sample. Variants not appearing in each of two independent cultures were removed, as were variants also appearing in the isogenic wild-type strain, HME63.

## Strain mutagenesis assay

To determine the mutation frequency of each strain, a rifampicin resistant assay was performed as previously described [82]. To test the base mutagenesis rate, fresh overnight cultures were subcultured 1:1000 in LB and allowed to outgrow for 24 hours at 32˚C before spreading onto plates containing 50 μg/mL rifampicin (Rif) (Thermo Fisher, Waltham, MA). Overnight cultures were subcultured in LB until $OD_{600}$ ~ 0.4, then cultures were pelleted and washed with PBS before resuspension in LB and grown overnight with shaking. Identical aliquots were plated onto $Rif^+$ and diluted and plated onto Rif- plates for colony-forming unit (CFU) controls. All incubation and shaking steps were performed at 32˚C. Mutation frequency was calculated as the ratio of mutants to total CFUs as follows:

$$\frac{A}{B \times 10^8} = mutation\ frequency \tag{6}$$

where *A* is the number of mutant CFUs (colonies on the Rif+ plate), *B* is the number of total CFUs (colonies on the Rif- plate), and $10^8$ is the dilution factor for *B*.

## SOS induction assay

Mutant and parental strains were transformed with the plasmid pEAW915, which contains SuperGlo GFP (Qbiogene) under the control of the *E. coli* recN promoter, in the plasmid pACYC184 [48]. Successfully transformed cells were selected by Kanamycin resistance. SOS fluorescence curves were recorded by diluting overnight clonal cultures 1:1000 in LB or LB with 0.001 μg/mL MMC (Thermo Fisher, Waltham, MA) and aliquoting 250 μL into a white clear-bottomed 96-well plate (Corning). The cultures were incubated at 32˚C with aeration at 225 RPM and the both the $OD_{600}$ and fluorescence (474 nm excitation / 509 nm emission) were recorded at 30-minute intervals using a Varioskan Lux multi-mode microplate reader equipped with SkanIt 5.0 software (Thermo Scientific, Waltham, MA). Specific fluorescence was determined by dividing the fluorescence by the absorbance to control for population density using the following equation:

$$Spec\ Fluor = \frac{fluoresence\ (509nm)}{absorbance\ (OD_{600})} \tag{7}$$

Data was processed using KaleidaGraph (Synergy Software, v.4.5.3) and fit to **Eq 2** for quantification of the upper asymptote.

Alternatively, LexA cleavage was monitored directly by western blot. Overnight cultures were diluted 1:1000 in LB and grown with aeration until exponential phase ($OD_{600}$ ~0.4–0.6); then 50 mL samples were pelleted and resuspended in 500 μL chemical lysis buffer at 4˚C. Samples were then sonicated 3 times at 80% for 10 seconds on ice. BCA protein assay (Thermo Fisher, Waltham, MA) was used to determine total protein concentration, and 50 ng total protein per sample was loaded prior to electrophoresing samples in a 15% SDS-acrylamide gel. Protein was transferred to a PVDF membrane (EDM Millipore, Burlington, MA), blocked

with 2% BSA, and then probed with rabbit-α-LexA (1:500) (EMD Millipore, Burlington, MA, cat# 06719) or rabbit-α-tau (1:500) (gift from Charles McHenry) [83] for 1 hour with rocking at 23˚C. After three 5-minute washes with TBST, blot was probed with secondary goat anti-rabbit-Alexa Fluor647 antibody (1:1,000) (Life Sciences, Invitrogen, Carlsbad, CA, cat# A27040) with rocking for 1 hour at 23˚C, then imaged using a Typhoon FLA9000 Imaging System (Cytiva, Marlborough, MA).

## Terminal deoxyribonucleotide transferase-mediated dUTP nick end labeling (TUNEL) assay

For microscopic imaging of DNA breaks, exponential growth cultures were harvested at OD ~ 0.3 by pelleting and washing PBS, then fixed in 1 mL of ice cold formaldehyde solution (4% paraformaldehyde in 1x PBS) as described [84]. Cells were incubated in fixing solution for 30 minutes at room temperature, pelleted, and washed with PBS. As a positive control, an exponential growth culture of the parental strain was exposed to 0.01 μg/ml MMC (Thermo Fisher, Waltham, MA) for 60 minutes prior to harvesting. As a method control, an exponential growth culture of parental strain was exposed to 10 mM HU (Thermo Fisher, Waltham, MA) for 4 hours prior to harvesting. After fixation, cells were resuspended in 500 μL ice cold permeabilization solution (0.1% Triton X-100 and 0.1% sodium citrate) and incubated on ice for 2 minutes. Cells were again pelleted, washed, resuspended in PBS, and stored at 4˚C overnight.

dUTP was added to DNA ends by pelleting stored cells and resuspending in 100 μL of elongation buffer (1X terminal deoxytransferase [TdT] buffer, 2.5 mM $CoCl_2$, 0.1 mM BrdU, 5 U of TdT [Thermo Fisher, Waltham, MA]) and incubating at 37˚C for 60 min. After elongation, cells were pelleted, washed with PBS, and then resuspended in blocking solution (4% BSA in 1X TBST) for 30 minutes at room temperature. To fluorescently label BrdU labelled ends, blocked cells were pelleted, washed with blocking solution, and resuspended in 100 μL of primary antibody solution (1:100 mouse-α-BrdU [BD Bioscience, Franklin Lakes, NJ] in TBST with 4% BSA) for 60 minutes at room temperature. Afterwards, cells were pelleted and washed with blocking solution, resuspended in 100 μL of secondary antibody (1:500 α-mouse IgG-Alexa647 [Thermo Fisher, Waltham, MA] in TBST with 4% BSA), and incubated in the dark for 60 minutes at room temperature. Cells were pelleted again, washed once with TBST, then washed with and resuspended in PBS before mounting using DABCO-DAPI solution. Slides were stored at 4˚C overnight and then imaged by epifluorescence microscopy as described above.

## Supporting information

**S1 Table. Results of Genome Sequencing of the *dnaB:mut* strains.**
(XLSX)

**S2 Table. Oligonucleotides.**
(XLSX)

**S3 Table. Strains.**
(XLSX)

**S1 Fig. Constricted Crystal of *E. coli* DnaB with mutants mapped.** Crystal structure of dilated DnaB (PDB: 2r6a), constricted cracked DnaB (lockwasher, PDB: 6qem), and constricted DnaB (PDB: 3bgw) with mutated residues highlighted: R74A in red, R164A in orange, K180A in green, and R328/9A in blue. The linear protein map (top) shows the location of each mutation relative to functional domains.
(TIF)

**S2 Fig. Duplex translocation assay unwinding controls.** DNA duplex unwinding by WT DnaB (500 nM monomers) to ensure (**A**) translocation over DNA180 without displacement and (**B**) unwinding and separation of the DNA180/181 fork. 20 nM of annealed DNA substrate was incubated with 500 nM DnaB (monomers) for 5 minutes at 37°C, initiated with 1 mM ATP and 150 nM respective unlabeled trap strand, and then EDTA quenched with 150 nM trap strand at indicated time points. %Unwound is indicated below each lane.
(TIF)

**S3 Fig. CRISPR-Cas9 recombineering to create *dnaB:muts*.** (**A**) CRISPR-Cas9 recombineering using the dual-plasmid system. A target gRNA for the desired mutation site on the *dnaB* gene was inserted into the *BsaI* cloning sites of pCRISPR, before electroporating both the pCRISPR plasmid and the recombination DNA oligonucleotide, engineered to contain the mutation, a novel restriction enzyme site for screening, and a point mutation to disrupt the PAM (5'-NGG) sequence. (**B**) Restriction digest gels for each of the engineered *dnaB* mutants showing successful digest at the novel restriction site and confirming *dnaB* gene mutation for several colonies. The frequencies of positively edited *dnaB* were 91% for R74A, 83% for R164A, 42% for K180A, and 69% for R328/9A.
(TIF)

**S4 Fig. Broader view of stationary and exponential cells and corresponding quantification of *dnaB:mut* cell length.** (**A**) Exponentially growing cells or overnight stationary phase cultures were imaged by microscopy, and (**B**) stational phase cell lengths were measured by blinded visual quantification. Average cell length is represented by the black bar in the middle of the data set. Average length in order from left to right: $2.0 \pm 0.5$ µm, $1.9 \pm 0.6$ µm, $2.6 \pm 0.9$ µm, $2.2 \pm 0.8$ µm, and $1.9 \pm 0.7$ µm. $n \geq 400$ events. Black bars above graph indicate statistically significant differences, where p-values are $^* < 0.05$.
(TIF)

**S5 Fig. *dnaB:mut* strains show changes in cell size and complexity populations.** Cell cultures exposed to rifampicin and cephalexin were then analyzed by FACS. The forward (FSC) and side scatter (SSC) gains were set based on the parental strain, and then scatter plot data was collected for all strains, n = 10,000 events. Dense populations of cells are indicated with red, moderate with green, and minor or diffuse population with blue. The parental strain has a single cell population that is primarily green. *dnaB*:R74A has a single population that is the most concentrated of all the strains (including parental), signified by strong red signal. *dnaB*:R164A, *dnaB*:K180A, and *dnaB*:R328/9A are significantly more diffuse than the parental and *dnaB*:R74A.
(TIF)

**S6 Fig. Rifampicin 'run-out' FACS reveals unique concentrations of DNA per cell size for *dnaB:mut* strains.** FITC versus FSC (forward scatter) plot of cell cultures exposed to rifampicin and cephalexin analyzed by FACS. FITC indicates the chromatin staining intensity, and FSC indicates cell size. Dense populations of cells are indicated with red, moderate with green, and minor or diffuse population with blue. Fixed yellow box is intended to highlight shift in the location of populations. The parental strain has two major cell populations, with cell size increasing with chromatin. *dnaB*:R74A is shifted to the left, with small cells containing large amounts of chromatin. *dnaB*:R164A has three elongated FITC populations, meaning that a single concentration of chromatin is contained within a wide range of cell sizes. The cell populations of *dnaB*:K180A have lost definition, diffusing into one another. *dnaB*:R328/9A only has a single large population near the top of the FITC axis, indicating that this strain contains almost

exclusively varied cell sizes with large amounts of chromatin.
(TIF)

**S7 Fig. Run-out of parental (*dnaB:WT*) cells have major peaks for 2 and 4 chromosomes.** A control experiment measuring chromosome density for the parental strain *dnaB:WT* and a single-chromosome strain, *dnaA46*(Ts). Chromosome density was measured by flow cytometry (FACS) in log phase rifampicin 'run-out' cultures stained with Sytox Green (n = 10,000 events). *dnaA46* was grown at the nonpermissive temperature (42˚C) to synchronize the culture before FACS analysis as described in the Materials and Methods. Chromosome integers are indicated at the top of the graph.
(TIF)

**S8 Fig. Size exclusion chromatography loading assay for the DnaB mutants.** DnaB **(A)** R74A, **(B)** R164A, **(C)** K180A, and **(D)** R328/9A were preincubated with DnaC, M13, and ATP before injecting onto a preequilibrated S200 10/30 size exclusion column according to the Materials and Methods. Example chromatogram and associated SDS-PAGE (Coomassie) and agarose (SYBR-Gold) gels used to monitor loaded DnaB and Free DnaB areas.
(TIF)

**S9 Fig. *dnaB:muts* frequency and characterization of genomic mutations.** Classification and characterization of transition (Ts) and transversion (Tv) genomic mutations in the *dnaB:muts* strains.
(TIF)

**S10 Fig. FACS data for *dnaB:mut* TUNEL assay.** Flow cytometry data of log phase cells plotted to show the relationship between DNA breaks and total amount of DNA. In these smoothed density plots, red represents concentrated cell populations, and dark blue represents highly diffuse cell populations. The grid is present to highlight changes in the size or location of cell populations. Small overall areas indicate concentrated populations that have a uniform and consistent distribution of DNA breaks (as that seen for parent + MMC). Noticeable tailing relative to the parental strain represents populations of cells that have increased DNA and DNA damage (as seen for *dnaB:R74A*). Y-axis shifts indicate changes in DNA repair or damage sensitivity; while X-axis shifts indicate changes in the amount of chromatin.
(TIF)

## Acknowledgments

Special thanks to Michael Cox and Elizabeth Wood (UW-Madison) for helpful advice and for providing bacterial strains and plasmids, and to the Court lab for generously providing strains for genome editing. We thank Preston Jones for aiding in developing the bacterial editing of *dnaB*. We also thank Susan Rosenberg, Jun Xia, and the rest of her lab (BCM) for helpful discussion and providing us with additional bacterial strains. We acknowledge the Baylor Molecular Bioscience Center (MBC) and the Center for Microscopy and Imaging (CMI) for providing instrumentation and resources aiding this project. We thank Oyindamola Adefisayo (Cornell Medicine) and all members of the Trakselis laboratory for productive conversations and insight.

## Author Contributions

**Conceptualization:** Megan S. Behrmann, Michael A. Trakselis.

**Formal analysis:** Megan S. Behrmann, Himasha M. Perera, Joy M. Hoang, Bryan J. Visser, David Bates, Michael A. Trakselis.

**Funding acquisition:** Michael A. Trakselis.

**Investigation:** Megan S. Behrmann, Himasha M. Perera, Joy M. Hoang, Trisha A. Venkat, Bryan J. Visser, David Bates, Michael A. Trakselis.

**Methodology:** Megan S. Behrmann, Himasha M. Perera, Michael A. Trakselis.

**Project administration:** Michael A. Trakselis.

**Resources:** Michael A. Trakselis.

**Supervision:** Michael A. Trakselis.

**Validation:** Megan S. Behrmann, Michael A. Trakselis.

**Visualization:** Megan S. Behrmann, Michael A. Trakselis.

**Writing – original draft:** Megan S. Behrmann, Michael A. Trakselis.

**Writing – review & editing:** Megan S. Behrmann, Michael A. Trakselis.

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
