## [Decision Letter · Decision Letter 0]

17 Jun 2021

Dear Dr Trakselis,

Thank you very much for submitting your Research Article entitled 'Targeted chromosomal Escherichia coli:dnaB exterior surface residues regulate DNA helicase behavior to maintain genomic stability and organismal fitness' to PLOS Genetics.

The manuscript was fully evaluated at the editorial level and by independent peer reviewers. The reviewers appreciated the attention to an important problem, but raised some substantial concerns about the current manuscript. Based on the reviews, we will not be able to accept this version of the manuscript, but we would be willing to review a much-revised version. We cannot, of course, promise publication at that time.

Should you decide to revise the manuscript for further consideration here, your revisions should address the specific points made by each reviewer. In particular, it will be important for you to provide further clarification to the data presented, since 2 of the reviewers thought that the phenotypes described do not lead to a clear interpretation of the effect of the DnaB mutants in the cell. The reviewers suggest ways to improve this aspect of the paper. We will also require a detailed list of your responses to the review comments and a description of the changes you have made in the manuscript.

If you decide to revise the manuscript for further consideration at PLOS Genetics, please aim to resubmit within the next 60 days, unless it will take extra time to address the concerns of the reviewers, in which case we would appreciate an expected resubmission date by email to plosgenetics@plos.org.

[LINK]

We are sorry that we cannot be more positive about your manuscript at this stage. Please do not hesitate to contact us if you have any concerns or questions.

Yours sincerely,

Rodrigo Reyes Lamothe

Guest Editor

PLOS Genetics

Josep Casadesús

Section Editor: Prokaryotic Genetics

PLOS Genetics

Reviewer's Responses to Questions

**Comments to the Authors:**

Reviewer #1: Summary: In this manuscript Behrmann et al., address the importance of helicase regulation in replication and genome maintenance. DnaB steric exclusion and wrapping (SEW) mutants result in faster unwinding of DNA in vitro. The authors use four mutants on this interface to ask what the impact of DnaB mis-regulation is on DNA replication and genome stability in vivo. They use an in vitro duplexed fluorescence translocation fork assay to first assess which conformation the SEW mutants adopt and conclude that these are likely to be more locked in the constricted state (in line with the increased fork speed measured in vitro). However, it is apparent that the four mutants behave somewhat distinctly: the authors introduce the mutants in vivo to assess the impact on cellular fitness, replication and genome stability. They find that strains carrying the mutants are compromised in growth and display heterogenous cell size distributions. Indeed, each mutant also impacts chromosome copy number and ori:ter ratio in distinct manners with the R328/9A mutant having the most pronounced effect on replication completion. Counterintuitively, this mutant is significantly compromised in loading on ssDNA. Given the impact of the mutants on cell growth and replication, the authors hypothesize an impact of the mutants on genome integrity. They find that two mutants resulted in increased mutagenesis. On the other hand, SOS response was activated in only R328/9A. However, unlike the SOS activation, all mutants showed likely increase in DSBs. The authors conclude that DnaB regulation via SEW interface is important for genome stability and efficient replication.

Strengths: Behrmann et al., ask an important question about helicase regulation and the coupling of helicase progression with the rest of the replisome in steady state growth conditions. They take a nice approach of testing the impact of well-characterized in vitro mutants on in vivo function. Their study reveals the significance of this interface on helicase regulation and the impact of the same on genome maintenance.

Limitations: While the approaches used by the authors provide support to the idea that helicase regulation is important, the results do not provide a definitive understanding of how mis-regulation may impact DNA replication and / or genome stability. This is in part due to the distinct phenotypes of the mutants. Thus, it remains unclear as to how exactly the regulation of the constriction-dilation states affects DNA replication and why this might in turn impact genome stability. For example, are the effects of the mutants distinct due to differences in the lengths of ssDNA gaps on the chromosome? It is possible that the genome instability phenotype is caused due to RecA mis-loading on long ssDNA tracts in some cases. Similarly, it is unclear as to how and why DSBs are generated each of the mutant contexts. Indeed, answers to some of these open questions may be outside the scope of the present study.

Major comments: I have the following specific comments that the authors should address in order to strengthen/ clarify observations in the present study.

1. Authors should combine the dnaB mutants with deletions of recA and separately recO (or recR) to assess for synthetic lethality or if the absence of RecA loading can alleviate the phenotypes observed in the mutant(s).

2. The assessment of the SOS induction is important, but the experimental design used by the authors is complicated. Why is there a need to pre-induce with damage prior to SOS measurement? The cell size heterogeneity and presence of breaks would strongly suggest that authors should be able to measure SOS induction levels without the need for a complicated experimental setup. Can the authors provide measurement of promoter activity in log phase cultures for mutants and wild type from discrete time points without such damage induction (time course is not required for this experiment). This should be corroborated with a Western Blot for RecA/ LexA.

3. The experiment carried out by the authors to determine break formation is relevant and the assay would be robust in the context of DNA damage treatment. However, in context of the dnaB mutants, I wonder whether the lysis technique would impact ssDNA tracts that may be labile to damage. The authors should provide a hydroxyurea-treated control for comparison, where TUNEL labeling should likely not increase more than wildtype (no damage cells).

4. The discussion section would benefit from significant rewriting to compare and contrast, with clarity, the various observations of the mutant phenotypes and the possible mechanisms resulting in the same. The authors should also clearly comment on the extent of ssDNA gaps each mutant might generate and if the same can contribute to the distinct phenotypes observed. Currently, as it is presented, it is unclear as to how and why the perturbations in DnaB toggling between constricted and dilated states results in the myriad phenotypes observed and the model figure does not highlight the impact of the specific mutants in this context.

Minor comment (1): L311 ‘poles’ instead of ‘polls’

Minor comment (2): Figure S6 is referred to in the manuscript after Fig. S7. Perhaps the order of these figures can be changed.

Reviewer #2: In this study Behrmann and co-workers describe experiments in E. coli that investigate how DnaB helicase mutants that favour the constricted conformation impact cell growth and genomic stability. The authors show that if a short duplex stretch is located in front of a frayed substrate, the mutants carrying such mutations either have a much-reduced ability to unwind the substrate or cannot unwind the substrate at all, in line with the idea that the constricted conformation for the DnaB helicase is favoured by the mutants. The authors then go on to show that essentially all of the mutants impact the doubling time, and at least some mutants result in the formation of elongated cells. By using direct competition experiments the authors show that mutant cultures are outcompeted by strains carrying the wild type allele. Run-out experiments with rifampicin show many of the mutants accumulate either more chromosome equivalents or, in addition, complex duplication intermediates. In line with the reduced growth rate and the elongated cells, some of the mutants show an increased mutation frequency, while for the induction of the SOS response more mixed results were found. Finally, the authors directly visualise dsDNA breaks via a bacterial TUNNEL assay and present a molecular model of when the two different states of the DnaB helicase are of particular importance.

Overall, I enjoyed studying the data presented. The authors have produced a large body of work that is well presented and in a logical order, and the questions asked make a lot of sense. However, in its current format the study is not yet scientifically sound, and additional work needs to be done before the work is ready for publication. I hope the comments below will help to improve the work.

Major points

1. In the first paragraph of the Introduction I was slightly lost what the authors actually meant by the term “helicase regulation”. This is then explained as part of the second paragraph, but I wondered whether the integration of what the authors mean, precisely, could be clarified earlier.

2. Line 103, this section can be improved. For example, an increased SOS response does not necessarily mean increased genomic instability. The authors should take care to be precise about their phrasing, a problem that also applies to a number of other sections (see below).

3. Line 130, in line with the comment above, the statement “… in this duplex translocation assay, both K180A and R328/9A mutants are unable to translocate over duplex DNA and therefore maintain a static fully constricted conformation” is not correct, as it is an implication. The assay is indirect, and no direct evidence is presented that the helicase is indeed in the constricted state. I agree with the authors that this is indeed a likely explanation, but the phrasing still needs to be accurate. This also applies to other sections of the manuscript (Line 134 to begin with), including the legend for Figure 1 (Line 682).

4. For this particular set of experiments – can the authors rule out that the small unlabelled duplex stretch is not unwound? The entire interpretation is based on this fact. It would be very important to clarify this point, either by clarifying in the text and citing the right source if this was done before, or by showing the relevant data.

5. Line 145, why are the authors using the rather unusual temperature of 32°C? Obviously 37°C is the standard temperature used, and 30°C is normally used if temperature-sensitive alleles are used.

6. There is a problem with the OD measurements, as these are influenced by cell size, and the authors actually demonstrate that, for some mutants, cells are elongated. The same OD for two cultures with different cell sizes means different viable titres. In fact, in Figure 3 the authors show the decline of all mutant cultures relative to cells carrying the wild type allele. Especially for R328/9A how is this possible if the growth rate is not so terribly dissimilar to wild type cells? R164A and K180A are vastly different, but disappear with the same kinetics. This does not make much sense. It could be in part because the cell sizes differ, and for this reason the comparison of the OD readings is not working particularly well, but that would need to be established.

Also, the authors should state the more commonly-used doubling time here. The graphs they show highlight steady growth, but the doubling time is very low in comparison to other experiments, with a doubling time of hours, rather than the usual 20 min for E. coli. This will be, in part, caused by the lower temperature and also by the fact that agitation and therefore aeration is naturally limited in plate readers. Still, in our lab cultures grow in 60 min from 0.2 to 0.8 at 37°C, while they take about 5 h in their set-up. It needs to be highlighted that the growth conditions are rather sub-optimal here.

7. Do the authors mean “nucleoid”, rather than “nucleotide”? Needs correcting throughout.

8. Line 226, the statement here needs either a reference or data. In fact, the same effect would occur if cells have a segregation defect or accumulate replication intermediates that cannot be properly resolved, which would be in line with the observed cell elongation.

9. Line 251, if R328/9A has a significantly reduced loading efficiency how come it is growing so relatively fast?

10. Line 264, all experiments so far were done in rich medium as far as I can see. Now the authors suddenly switch to M9. Why the change? Rif rates can be established perfectly fine in rich medium as well.

11. Line 266, it is extremely important that the authors use precise language here. What they have established is *not* a mutation rate. Instead, they have measured frequencies, for which they calculated the arithmetic mean (see below). I believe in the M&M section this was correctly described; it cannot be called a rate here. Also, for a frequency the authors cannot, and should not, calculate the arithmetic mean. Mutation rates are established by a rather elaborate fluctuation analysis, which requires the growth of multiple parallel cultures (11 or more). I am not saying that the authors need to establish rates here, but when doing a fluctuation analysis, it becomes obvious that in some cultures the first event is, by chance, happening early, leading to a culture with very high numbers of Rif-resistant cells. In a fluctuation analysis these cultures will be deliberately discarded. However, by calculating the mean the authors include what can be a vast variability which is *not* representative for the actual mutation rate in their calculations. It would be much better to show individual data points here, rather than an average.

12. Line 289, why was MMC included here? And why are experiments without MMC not shown?

13. Line 304, the authors use the term “Log phase cultures” here and elsewhere. While very common, it is not good practise to use the term “logarithmic growth”. What is that supposed to be? While understandable for most readers, the term “exponential growth” is a much better choice and should be used throughout the manuscript.

14. Line 484, the composition of “LB” needs to be defined here, as there are at least three different recipes which differ (mostly) in salt content.

15. Line 521, the procedure here needs to be clarified. How exactly were the cultures treated? Were cells passaged to refresh growth? And if so when were the dilution steps done?

Minor points

Line 40, an “all” is missing before “Domains”, and the latter should be lowercase.

Line 165, “Large” should be lowercase.

Line 311, I believe the authors mean “poles”, rather than “polls”.

Line 348, I believe this statement needs a reference.

Line 359, what is “G. ste” supposed to mean?

Line 735, the grammar is not quite right here.

Line 736, “preequilibrated” needs a hyphen

Reviewer #3: In this manuscript the authors follow on from their previous work to try to further characterize a set of surface residue mutants of DnaB. In previous work it was suggested that these mutants affect the interaction of the excluded strand of ssDNA with the surface of DnaB leading to a loss of control of helicase activity.

In this work the authors suggest that these mutants alter the ability of the N-terminal domain of DnaB to undergo a transition from a constricted to dilated state, based on the inability of the mutant DnaBs to pass over a section of dsDNA.

The work then looks at in vivo effects of introducing these mutants into E. coli cells by a number of assays. The phenotypes are quite complex and not clear-cut, making interpretation tricky. However, it is clear that the mutations are deleterious, lead to chromosomal abnormalities, increased mutation rates, increased DNA content in cells and chromosome segregation problems.

One suggestion for an improvement would be to have reference to Figure S8 much earlier, so that it is more obvious where the mutants that are used in the study map to. I found myself wanting a figure like this from the introduction but only discovered it was present by the discussion.

Specific comments:

Heading: “Introuction” should be “Introduction”

Introduction, line 55: is “proximal” the right word? In this case do you mean central or vital to the process, rather than adjacent to?

Line 58 should be “unwinding by…” rather than “unwinding of”

Line 66 “helicase mechanism dysregulation”- could be just “helicase dysregulation”

Line 77-79 “This regulation may be important in vivo to limit separation of DnaB from the replisome, which may occur during Okazaki fragment priming or during helicase-polymerase decoupling [22, 23].” Are these the correct references here? The Mangiameli et al paper discusses helicase transcription conflicts but I don’t recall any mention of uncoupling of helicase and polymerases or primase? Similarly in the Nature paper I don’t recall these points being discussed?

Making site-specific mutants: one problem in making mutants in an essential gene is the large selection pressure for suppressor mutants in alleles which are deleterious. It would be good to know how easily each SEW mutant could be transduced into a wt strain (or alternatively to see the whole genome sequence of the mutants compared to their parental strain to detect suppressors). Of course, the CRISPR approach used does not generate an associated selectable marker, so transduction would not be easy.

One possible indication of whether each mutation is deleterious is the relative efficiency that changes were detected in the dnaB gene following the CRISPR plasmid induction. Was each mutant detected with the same efficiency, or were some harder to make than others?

One concern is that, in vitro, constricted mutants did not show lagging strand synthesis (Monachino et al. 2020). A downregulation of lagging strand synthesis in vivo could account for the observed growth defect. However, if serious, then it could also lead to compensatory mutations in other genes to overcome this pressure. This possibility (compensatory secondary mutations) should be discussed, or evidence provided for why this is not a concern.

Growth rate experiments: what do the growth rate calculations mean, because visually they do not make much sense when you look at the exponential phase of the growth cycle: for example the wt goes from an OD of 0.2 to 0.4 in about an hour (i.e growth rate of 1 per hour?). It then goes from 0.4 to 0.8 in just over 2 hours. How does the growth rate come out to be 0.18? It does not seem to match with the actual doubling time of the cultures shown at exponential growth, but is designed to account for an increased lag phase as well? Perhaps a clearer explanation of the single growth rate parameter quoted would make it easier for readers to interpret the data.

The micrographs in Fig. 2 (even with the zoomed inserts) are too small and low quality to properly make out necessary details. Do the filaments have segregated nucleioids or not? This would be very interesting to know.

Line 196 –“least” rather than “lease”

Line 227: “route causes”- I think you mean root causes.

Ori:ter ratios. While this is generally supportive of a defect in these cells, it is likely that a delay in cell division due to some filamentation can cause these changes. To my mind, an ori:ter ratio of 3 in cells is an average of recently divided cells which have a ratio of 2:1 and pre-division cells which have an average of 4:1. If division is delayed then there are more 4:1 cells, possibly even 8:1 and fewer 2:1. The altered ratios in the mutants may just reflect delayed division?

What happens with exponentially growing cells by FACS? In wt cells you usually see a signal something like this: a peak at 2 chromosome copy number then an exponential decrease towards 4 copies per cell. If the mutants have problems with replisome stalling then there might not be the exponential decrease seen in wt. It would be interesting to see if introducing a mutation that destabilised RNA polymerase (eg rpoB*35) increased the fitness of these mutants, or if the fitness loss is not related directly to replisome collisions/collapse.

SOS induction and BrdU incorporation: it would also be interesting to see the relative viability/fitness of a recA mutant in the dnaB mutant backgrounds.

Line 336-338 and following sentences: “DnaG, which favors the dilated state of DnaB, has been shown to limit replisome progression and generate pausing events [47], consistent with a model where the helicase constricts”. It has been shown that in the presence of the clamp loader complex there is no pausing required for DnaG action and that there is likely a spiral, non-planar N terminus state (Monachino et al., 2020)

Line 379-80: “ori and/or ter regions of the chromosome, which migrate to the ends of the cell during chromosome segregation [52-54” Is this is true? In wt E. coli the ori region starts at midcell and then migrates to the ¼ and ¾ positions. The terminus region can be polar then migrates to midcell for duplication and remains there prior to segregation (for example see Wang, Possoz and Sherratt, 2005).

Line 419: would “harmony” be better than “symphony” here?

Line 570: 0.5 μL each of 100 μM forward and reverse primers. Is this correct- most often protocols would use 0.5 μL of 10 μM primer.

Figure S1: part A: the lower-right red arrowhead, showing the BsaI cleavage site is one base out from where it should be. It should cleave after AAAA (4bp overhang).

Fig S3. The circled small populations in the last 3 mutants- it is not convincing that these are real cells rather than some kind of background. In microscopy, were any small or mini cells (no DNA content) seen that could account for these populations (using light microscopy rather than the DAPI stain)?

**Have all data underlying the figures and results presented in the manuscript been provided?**

Reviewer #1: None

Reviewer #2: Yes

Reviewer #3: Yes

PLOS authors have the option to publish the peer review history of their article (what does this mean?). If published, this will include your full peer review and any attached files.

Reviewer #1: No

Reviewer #2: No

Reviewer #3: **Yes: **Ian Grainge

---

## [Decision Letter · Decision Letter 1]

18 Oct 2021

Dear Dr Trakselis,

We are pleased to inform you that your manuscript entitled "Targeted chromosomal Escherichia coli:dnaB exterior surface residues regulate DNA helicase behavior to maintain genomic stability and organismal fitness" has been editorially accepted for publication in PLOS Genetics. Congratulations!

Yours sincerely,

Rodrigo Reyes Lamothe

Guest Editor

PLOS Genetics

Josep Casadesús

Section Editor: Prokaryotic Genetics

PLOS Genetics

Comments from the reviewers (if applicable):

There was a consensus among the three reviewers that the revised manuscript improved significantly. I agree with them. There are still few minor points raised by reviewer 3, which include a suggestion on the presentation of your sequencing data, few clarifications on the methods section, and few typos and other mistakes. Clarifying the methods and correcting the text are particularly important and you should address them before submitting a final draft.

In addition, both reviewers 1 and 3 raised the point that the sequencing data seems not to be available. As suggested by them, the sequencing data should be uploaded to an online repository and the accession number should be included in the paper.

Reviewer's Responses to Questions

**Comments to the Authors:**

Reviewer #1: I think the authors have carried out a substantial revision of their originally submitted manuscript to address Review comments. While the results with regards to the SOS response are intriguing and needs additional experimentation, the authors have provided a reasonable explanation in the discussion (which is likely sufficient for the current manuscript). 

Authors should deposit sequencing data in an appropriate online repository and include an accession number in the manuscript.

Reviewer #2: The authors have presented a revision of an earlier draft, and there is little doubt that they have made substantial efforts to clarify and correct all points raised by the reviewers. I hope the authors agree that the process was constructive, as I find the revised version much improved. The descriptions are crips and to the point, the language is much more precise and accurate and the presentation, both in terms of text and images, is of a very high standard. The material was very interesting to begin with, but this has developed into a very nice paper.

Reviewer #3: I thanks the authors for improvements to the manuscript in line with the review comments. I think the paper is improved by your efforts. I still have a few questions and suggestions- the questions are mainly around the exact methodology employed in a few experiments that it is worth clearing up as it could affect the interpretation of the data. I also have a suggestion (which may yield nothing, or it may be of interest) which I will start off with:

Suggestion: since you have the whole genome sequencing data, can you do a marker frequency analysis using the relative read numbers across the chromosome, against the chromosome position (see for instance Rudolph, Upton et al 2013, Nature) to see if you get the expected “smooth” line decreasing from ori to ter on both replichores? Ideally each sample would be normalized against a stationary phase culture of the same strain but this may not matter- you might be able to just compare to read numbers of the wt strain at the same growth stage.

Do you see the over-representation of ori-proximal markers in your dnaB mutants that would be expected from your qPCR data? Do you see any under- or over- representation of the ter in R74A that might mean higher frequency of breakage, or displacement of the other leading strand when forks collide in the terminus (as may be implied by the BrdU incorporation in TUNEL)? Will this tell you something interesting about the replication in these cells above what you already know or point to where repair may be occurring?

Methodological questions:

Why is your recA deletion in a wt background so deleterious- 100-1000 fold fewer CFUs. Were ODs normalized before plating or did the recA strain grow more slowly from stationary for example? Are these plated directly from stationary?

The FACS data: I suspect that the dnaBR74A peaks, that do not perfectly line up with the peaks in the wt, are actually 4 and 8 chromosome content and not 5 and 9. The fact the peaks are so sharp makes me believe they are 4 and 8, whereas a loss of integer or multiple of 2 copy numbers would probably be a number of smaller peaks or a smear between peaks as seen with the other mutants. I have seen effect before using FACS where peaks between different strains don’t line up exactly. It seems as easier explanation than why this strain should have exactly 5 and 9 copies of the chromosome- how would that even happen? And if it could happen why does it happen to such an extent you exclusively see 5 and 9 not 6, or 7 or any other number?

This argument would be the same for the major peaks in the R164 strain too, but here there are minor peaks between the major ones.

According to the Methods, the glycerol stocks of your strains were made prior to outgrowth to remove the CRISPR plasmids. Are you sure that for all the TUNEL assays you used strains that you had confirmed were all plasmid free? The polar locacation of the BrdU signal in R74A looks very similar to the location of multicopy plasmids in E. coli cells.

With the increased mutagenesis rates of these cells, was the parental strain (which is also mutS) also grown out for a similar time as the dnaB derivatives to see how many mutations it accumulated over this time? The dnaB cells would be grown up to enable transformation with the CRISPR plasmids, expression of the CRISPR system, verification etc. Was the wt then also passaged this many times and the resulting strain compared to the original wt to see the number of mutations it accumulated over this amount of time? Certainly the rif mutation frequency would suggest elevated mutation rates, but it is worth being absolutely clear about how you carried out the sequencing experiments to determine the mutations.

Typos and minor adjustments:

Line 55 “effect” should be affect.

Line 60: I still don’t think proximal is the correct word here, I would just say central. Proximal to me means the closer, nearest or more central, end of something as in this definition: “proximal: situated nearer to the centre of the body or the point of attachment. "the proximal end of the forearm"” The process you are talking about is DNA replication, not the position of the helicase being proximal to the dsDNA at the replisome is it not?

Line 108 conformation not confirmation

Line 189: “SEW is important for strain efficacy and survival” What exactly is meant by strain efficacy? Again, I’m not sure it is the correct word to use to describe a strain. Is there another less confusing word that could be used- just fitness perhaps?

Line 254 “All dnaB mutants…”. Previous line specifies that the R74A was not significantly different from control, so change “All” to “3 of the 4 dnaB mutants…”

Line 277: loading of DnaB not “by DnaB”?

Line 343: just poles not terminal poles.

**Have all data underlying the figures and results presented in the manuscript been provided?**

Reviewer #1: Yes

Reviewer #2: Yes

Reviewer #3: **No: **Not sure if the WGS data has been made publicly available, or if it is required?

PLOS authors have the option to publish the peer review history of their article (what does this mean?). If published, this will include your full peer review and any attached files.

Reviewer #1: No

Reviewer #2: No

Reviewer #3: No

**Data Deposition**

http://datadryad.org/submit?journalID=pgenetics&manu=PGENETICS-D-21-00690R1

**Press Queries**

---

## [Editor Report · Acceptance letter]

8 Nov 2021

PGENETICS-D-21-00690R1 

Targeted chromosomal Escherichia coli:dnaB exterior surface residues regulate DNA helicase behavior to maintain genomic stability and organismal fitness 

Dear Dr Trakselis, 

We are pleased to inform you that your manuscript entitled "Targeted chromosomal Escherichia coli:dnaB exterior surface residues regulate DNA helicase behavior to maintain genomic stability and organismal fitness" has been formally accepted for publication in PLOS Genetics! Your manuscript is now with our production department and you will be notified of the publication date in due course.

With kind regards,

Katalin Szabo

PLOS Genetics

On behalf of:
